# Postnatal supplementation with alarmins S100a8/a9 ameliorates malnutrition-induced neonate enteropathy in mice

Lisa Perruzza[1,10,11] ✉, Julia Heckmann[2,11], Tanja Rezzonico Jost[1], Matteo Raneri[1], Simone Guglielmetti[3,4], Giorgio Gargari[4], Martina Palatella [1], Maike Willers [5], Beate Fehlhaber[5], Christopher Werlein [6], Thomas Vogl [7], Johannes Roth [7], Fabio Grassi [1,12] & Dorothee Viemann [2,5,8,9,12] ✉

Malnutrition is linked to 45% of global childhood mortality, however, the impact of maternal malnutrition on the child's health remains elusive. Previous studies suggested that maternal malnutrition does not affect breast milk composition. Yet, malnourished children often develop a so-called environmental enteropathy, assumed to be triggered by frequent pathogen uptake and unfavorable gut colonization. Here, we show in a murine model that maternal malnutrition induces a persistent inflammatory gut dysfunction in the offspring that establishes during nursing and does not recover after weaning onto standard diet. Early intestinal influx of neutrophils, impaired postnatal development of gut-regulatory functions, and expansion of *Enterobacteriaceae* were hallmarks of this enteropathy. This gut phenotype resembled those developing under deficient S100a8/a9-supply via breast milk, which is a known key factor for the postnatal development of gut homeostasis. We could confirm that S100a8/a9 is lacking in the breast milk of malnourished mothers and the offspring's intestine. Nutritional supply of S100a8 to neonates of malnourished mothers abrogated the aberrant development of gut mucosal immunity and microbiota colonization and protected them lifelong against severe enteric infections and non-infectious bowel diseases. S100a8 supplementation after birth might be a promising measure to counteract deleterious imprinting of gut immunity by maternal malnutrition.

Malnutrition persists as a heavy health burden worldwide[1,2] and might grow given the geoeconomic developments that deteriorate the food situation in many countries such as the COVID-19 pandemic and lasting territorial conflicts[3,4]. Protein-energy-malnutrition is the most prevalent form of undernutrition. It causes in children stunting and leaves them highly susceptible to infections, especially to enteric infections that account for more than 10% of child deaths[5–7]. However, the precise nature of the susceptibility to infections remains uncertain

[1]Institute for Research in Biomedicine, Faculty of Biomedical Sciences, Università della Svizzera Italiana (USI), Bellinzona, Switzerland. [2]Department of Pediatrics, University Hospital Würzburg, Würzburg, Germany. [3]Department of Biotechnology and Biosciences (BtBs), University of Milano-Bicocca, Milan, Italy. [4]Department of Food, Environmental and Nutritional Sciences (DeFENS), University of Milan, Milan, Italy. [5]Department of Pediatric Pneumology, Allergology and Neonatology, Hannover Medical School, Hannover, Germany. [6]Institute of Pathology, Hannover Medical School, Hannover, Germany. [7]Institute of Immunology, University of Münster, Münster, Germany. [8]Center for Infection Research, University Würzburg, Würzburg, Germany. [9]Cluster of Excellence RESIST (EXC 2155), Hannover Medical School, Hannover, Germany. [10]Present address: Humabs BioMed SA a Subsidiary of Vir Biotechnology Inc., Bellinzona, Switzerland. [11]These authors contributed equally: Lisa Perruzza, Julia Heckmann. [12]These authors jointly supervised this work: Fabio Grassi, Dorothee Viemann. ✉e-mail: lperruzza@vir.bio; Viemann_d@ukw.de

due to considerable gaps in current understanding of immune dysfunction in malnutrition[8].

A particular sequela of concern is a chronic inflammatory gut dysfunction called environmental enteropathy (EE) associated with malnutrition. EE is characterized by chronic gut inflammation, villus blunting and impaired barrier function leading to malabsorption that in turn further aggravates malnutrition, catching affected individuals in a vicious cycle[9–11]. Thus, EE is both an effect and a cause of malnutrition. Since EE is primarily diagnosed in patients living in poor and unsanitary conditions[9,12], frequent uptake of pathogens and altered intestinal colonization are discussed to induce EE[10,13–17]. In mouse models that start malnutrition after weaning, additional microbial challenges or intestinal injuries are required as co-impulses to induce EE[18,19]. However, translation of these findings into trials that improved hygienic conditions failed to promote intestinal health in malnourished (MN) children[20–22]. Thus, there is an urgent need to identify the major factors causing the enteropathy associated with malnutrition.

So far, little attention has been paid to the impact of malnutrition during the first weeks after birth. An important maternal contribution to child's health is breastfeeding[23–25]. Malnutrition has been claimed to have little effect on the composition of breast milk[2,26], while studies performed in disadvantaged communities hint to a role of insufficient breastfeeding in the development of EE[27,28]. Several components in breast milk modulate the gut immune and microbiota development in the offspring[23,25], e.g., the alarmins S100A8/A9, which are known key factors in the establishment of gut homeostasis in both murine and human neonates[29–31]. However, the role of specific breast milk components in the context of maternal malnutrition and the offspring's immune development has not been elucidated so far[8].

In this study, we established a murine model of maternal malnutrition to investigate whether and at what age EE manifests in an offspring raised under malnutrition conditions. Hallmarks of altered postnatal development of gut mucosal immunity and microbiota colonization in neonates fostered by MN mothers are revealed, aiding in understanding how the increased susceptibility to severe enteric infections evolves. We identified deficient S100a8/a9-priming of the neonatal gut as crucial pathogenetic factor of the enteropathy induced by maternal malnutrition and demonstrate that a single nutritional supply of S100a8 after birth protects long-term from it and related medical complications.

## Results

### Maternal malnutrition impairs the postnatal development of gut mucosal immunity

To investigate how maternal malnutrition modulates the development of gut mucosal innate immunity in neonates, we established a murine model of malnutrition during pregnancy and nursing period and performed longitudinal studies in the offspring at defined ages (d7 and d23) (Fig. 1a). To control for environmental variables known to influence mouse biology[32], mouse experiments were carried out at two different study sites. The calorie contents of the standard and malnutrition diets were similar (Supplementary Table 1) and body weights of MN and WN dams remained comparable until the termination of malnutrition (Supplementary Fig. 1a). The proportion of protein from calorie content was 3.9-fold lower in the malnutrition diet, while that of fat was also reduced (2.3-fold) and that of carbohydrates 1.5-fold higher compared to the standard diet (Supplementary Table 1). Interestingly, MN dams consumed more chow and calories per day than well-nourished (WN) dams (Supplementary Figs. 1b, c). Yet the mean energy uptake of MN dams by protein was still 3.0-fold reduced compared to WN dams, while that by fat remained 1.8-fold lower and that by carbohydrates 1.9-fold higher (Supplementary Figs. 1d–f). In the offspring, the induction of stunting and enteropathy was verified in d23 mice of MN compared to WN dams (in the following named MN

respective WN mice or pups) by demonstrating growth failure (Supplementary Figs. 2a–c), barrier dysfunction (Supplementary Figs. 2d, e) and chronic intestinal inflammation reflected by increased levels of lipocalin-2 (Lcn-2) in the stool (Fig. 1b) but not serum (Fig. 1c). The overall amount of lamina propria mononuclear cells (LPMCs) was only marginally reduced in the MN compared to the WN offspring, namely only at d23 in the small intestine (SI) but otherwise comparable (Fig. 1d). However, we observed a significant expansion of polymorphonuclear neutrophils (PMNs) in the SI and large intestine (LI) of MN pups at d7 (Figs. 1e, f). At d23, shortly after weaning onto standard diet, the abundance of PMN in MN mice had again declined to similar low levels as in WN mice (Fig. 1e). However, at that time point an altered lamina propria macrophage (LPMP) phenotype manifested in the MN offspring (Fig. 1g–j). The total amounts of LPMPs were largely comparable between WN and MN pups, only at d23 reduced in the LI of MN mice (Fig. 1g). Yet, we observed a significant expansion of chemokine (C-X3-C motif) receptor 1 low-expressing (Cx3cr1^low) LPMPs at d23 in both the SI and LI of MN mice (Figs. 1h, i), which is a proinflammatory LPMP subset of hematopoietic origin invading during inflammatory gut conditions[33]. In contrast, the tissue-resident Cx3cr1^high LPMP subset, which plays a key regulatory role in maintaining intestinal tissue homeostasis[29,33], was significantly diminished at d23 in the SI and LI of MN mice compared to WN mice (Fig. 1j). Moreover, Cx3cr1^high LPMPs from the LI of MN mice expressed significantly less MHC-II than those of WN mice (Fig. 1k), pointing to overall insufficient development of regulatory and antigen-presenting LPMP functions. Interestingly, at d7 the proportions of Cx3cr1^low LPMPs in the LI were lower (Fig. 1h) and that of Cx3cr1^high LPMPs higher (Fig. 1j) in MN than in WN pups, suggesting altered replacement of yolk sac-derived Cx3cr1^high LPMPs by Cx3cr1^low LPMPs, as Cx3cr1^low LPMPs migrate after birth physiologically into the colonic mucosa[29,34,35]. Thus, the cellular signs of intestinal inflammation developed most prominent in the LI. Colonic inflammation in the fetus appeared unlikely (Supplementary Figs. 3a–f), demonstrating that the early-life enteropathy developing under maternal malnutrition is less a result of maternal malnutrition during pregnancy but manifests primarily after birth during the nursing period.

Further evidence for maternal malnutrition impairing the establishment of gut homeostasis in the offspring was derived from gene expression studies of proinflammatory and regulatory key cytokines in d23 LMPCs. Increased expression of Tnf in the LI opposing strongly reduced expression of Il10 and Tgfb1 in the SI and LI of the MN offspring supported their overall inflammatory gut phenotype compared to the WN offspring (Fig. 1l).

Cx3cr1 along with Il-10 and Tgf-β are crucial factors that trigger the postnatal expansion of regulatory T cells (Tregs) in the gut mucosa[29,36–38]. In line with their reduced expression by LPMPs, Treg expansion was strongly reduced in the SI and LI of MN mice compared to WN mice (Fig. 1m), corroborating profound impairment of gut homeostasis.

Taken together, the data demonstrate that maternal malnutrition without additional pathogen challenge is sufficient to induce already in the neonatal period chronic intestinal inflammation in the offspring, impairing the development of regulatory gut mucosal functions and overall gut homeostasis.

### Maternal malnutrition-induced enteropathy paves the way for severe enteric infections in the offspring

Next, pups of WN and MN dams underwent two different enteric infection models with either Citrobacter rodentium (a murine mucosal pathogen modeling human infections with enteropathogenic and enterohaemorrhagic Escherichia coli[39]) or Salmonella enterica serovar Typhimurium (a common enteric pathogen in humans as well as mice[40,41]). In the C. rodentium model, pups were infected at d12 after birth and assessed 10 days later (Fig. 2a). In both experimental groups,

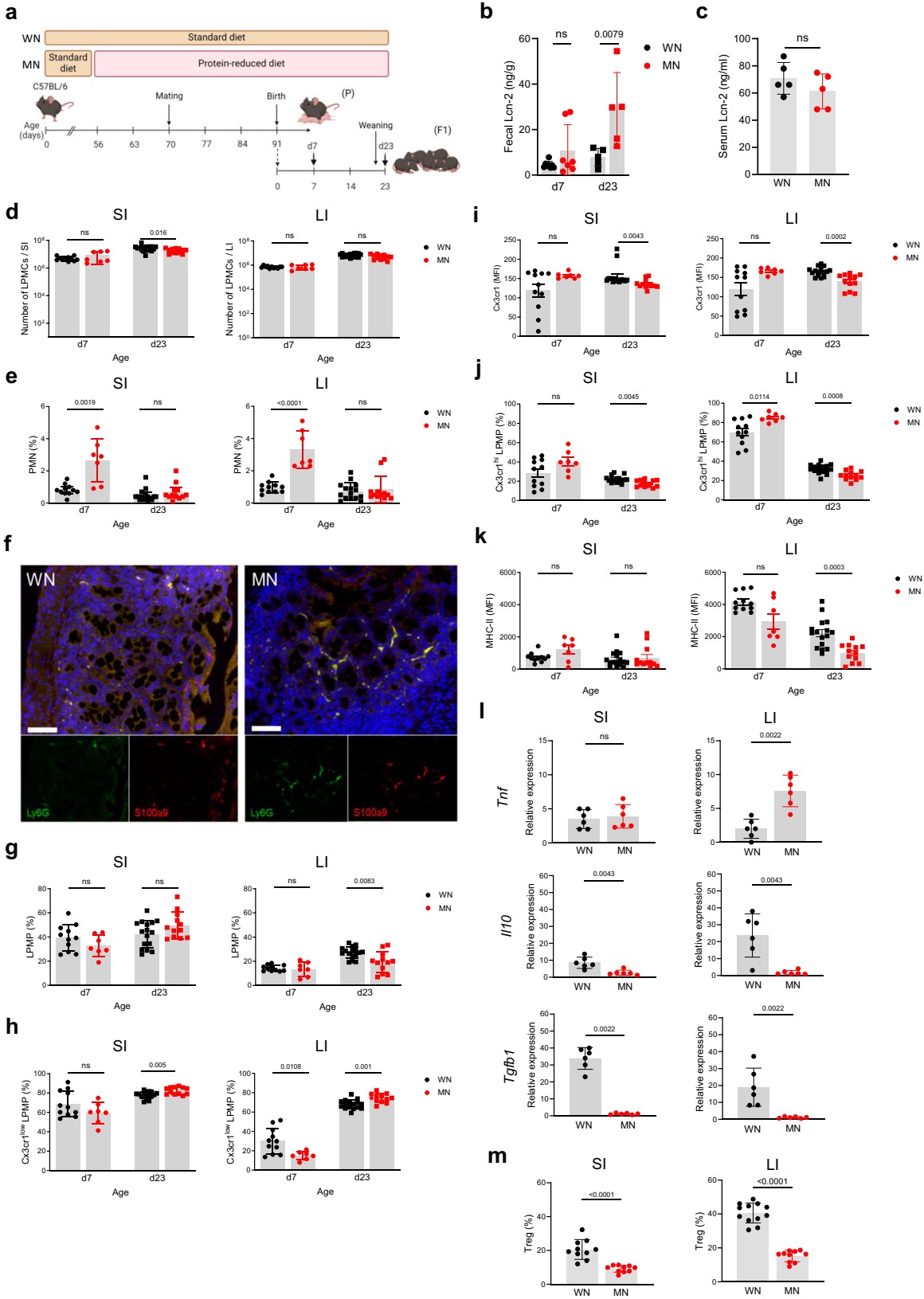

infection induced an equal increase of fecal Lcn-2 levels (Fig. 2b). However, MN pups ended up with higher fecal Lcn-2 levels than WN pups due to already increased levels at baseline, suggesting exacerbation of ongoing intestinal inflammation. This came with an exaggerated increase of systemic Lcn-2 levels in infected MN pups (Fig. 2b). The LI length was affected by the *C. rodentium* infection in both the WN

and MN groups compared to non-infected control mice. However, in the MN group, this effect again added up to the length restriction pre-established under malnutrition (Fig. 2c). The increased influx of PMNs into the LI mucosa (Fig. 2d) and strongly impaired intestinal barrier functions with increased translocation of *C. rodentium* into the spleens and the livers (Fig. 2e) underlined the more severe inflammatory gut

**Fig. 1 | The offspring of MN mothers develops an inflammatory gut mucosal phenotype lacking regulatory functions. a** Experimental setup. **b, c** Lcn-2 levels in fecal samples collected at d7 and d23(WN d7: *n* = 8, MN d7: *n* = 7; d23: *n* = 5 per group) (**b**) and serum samples collected at d23 after birth (*n* = 5 per group) (**c**) from the WN and MN offspring. **d** Number of LPMCs isolated from the SI and LI of WN and MN mice at d7 (WN: *n* = 11, MN: *n* = 7) and d23 (WN: *n* = 16, MN: *n* = 12). **e–k** Flow cytometric analysis of myeloid cells isolated from the SI and LI of WN and MN mice at d7 (WN: *n* = 11, MN: *n* = 7) and d23 (WN: *n* = 11, MN: *n* = 7). **e, g** Proportions of polymorphonuclear neutrophils (PMNs) (**e**) and lamina propria macrophages (LPMPs) (**g**) from LPMCs. **f** Representative images of LI tissue samples from d7 WN mice (left) and d7 MN mice (right) immunostained for Ly6G (green), S100a9 (red)

and nuclei (DAPI; blue). Lower panels, single color stainings of Ly6G and S100a9 respectively. Scale bars, 50 μm. **h** Proportion of Cx3cr1^low LPMPs from LPMPs. **i** Expression of Cx3cr1 on Cx3cr1^low LPMPs plotted as mean fluorescence intensity (MFI). **j** Proportion of Cx3cr1^hi LPMPs from LPMPs. **k** Expression of MHC-II on Cx3cr1^hi LPMPs plotted MFI. **l** Expression of indicated genes in LPMCs isolated from the SI and LI at d23 (*n* = 6 mice each group). **m** Proportion of Tregs from LMPCs isolated from the SI and LI at d23 (SI: *n* = 10 mice each group; LI: WN: *n* = 6 mice, MN: *n* = 8 mice). Bars represent means ± SEM. Exact *p*-values are displayed, *ns*, not significant (two-tailed MWU-tests). Panel a was created in BioRender under license number BioRender.com/c00z263.

condition in infected MN mice compared to WN mice. In line with the increased bacterial translocation, both intestinal and systemic total IgA levels were lower in MN than WN mice at baseline. Upon infection with *C. rodentium* systemic IgA was strongly induced in MN mice and became comparable to the levels in infected WN mice. Interestingly, intestinal IgA levels did not significantly change upon *C. rodentium* infection neither in WN nor MN mice (Fig. 2f).

In the *S. typhimurium* model, pups were infected at day 12 after birth and assessed 3 days later (Fig. 2g). The patterns of increased Lcn-2 levels (Fig. 2h) and stronger shortenings of LI lengths (Fig. 2i) upon *S. typhimurium* infection in MN mice than WN mice were similar to those observed after *C. rodentium* infection. Likewise, *S. typhimurium* infection of MN pups resulted in a significantly stronger influx of PMNs into the LI (Fig. 2j) as well as a more enhanced bacterial translocation into spleens and livers (Fig. 2k) than in WN pups. In the *S. typhimurium* model, the infection increased fecal IgA levels in both groups but in WN mice to higher levels than in MN mice, while systemic IgA levels increased strongly in MN mice but only slightly in WN mice (Fig. 2l).

These findings demonstrate in two different enteric infection models that maternal malnutrition predisposes the offspring to severe enteric infections as a result of an exacerbation of pre-established gut inflammation and loss of barrier functions.

## Maternal malnutrition alters the development of the gut microbiota and causes deficient S100a8/a9 supply

To determine whether the gut microbiota development is affected in neonates fostered by MN dams, we first performed culture-based analyzes in fecal samples collected at d7 and d23 (after weaning). Except for slightly lower levels of aerobic bacteria in d23 MN mice, the amount of total aerobic and anaerobic bacteria did not differ from those in WN mice. However, selective culturing revealed a strong, premature expansion of *Enterobacteriaceae* in the MN group already in the first week of life, while in WN mice *Enterobacteriaceae* expanded only later after weaning (Fig. 3a). To identify which members of the gut microbiota were specifically affected, we performed 16S rRNA gene profiling of d23 cecum contents. In comparison to the MN offspring, healthy WN mice showed an enriched abundance of members of the classes *Clostridia* (*Ruminococcaceae*, *Eubacteriaceae* or *Clostridiaceae*) and *Bacteroidia* (*Rikenellaceae*, *Prevotellaceae* or *Bacteroidaceae*). Oppositely, the MN offspring was characterized by an enrichment of the classes γ-*Proteobacteria* (especially *Enterobacteriaceae*) and *Bacilli* (e.g., *Streptococcaceae*, *Staphylococcaceae*) as well as the family *Lachnospiraceae* (Fig. 3b, Supplementary Data 1). This was supported by group-wise comparisons at the bacterial family level revealing significant differences in the relative abundances of *Enterobacteriaceae* (higher in MN *versus* WN) and *Clostridiaceae* (lower in MN *versus* WN) (Fig. 3c), highlighting the overexpansion of *Enterobacteriaceae* and the reduced abundance of *Clostridiaceae* as major malnutrition-associated microbiota alterations in our model.

We noticed that early intestinal overgrowth of *Enterobacteriaceae*, impaired development of regulatory LPMPs, and insufficient postnatal expansion of Tregs are also the striking features of altered gut

homeostasis following deficient S100a8/a9-priming of the neonatal gut[29]. Subsequent quantification in breast milk, the most important source of fecal S100a8/a9 for the infant[29,42], revealed significantly lower S100a8/a9 levels in MN compared to WN dams (Fig. 3d). Fecal S100a8/a9 levels in the offspring did not yet differ at birth (d1) but became lower in MN mice from d7 on and remained strongly decreased compared to WN mice even at d23 after weaning. S100a8/a9 deficiency of MN mice at d7 might have been mitigated by the infiltrating PMNs, which are well-known producers of S100a8/a9 (Figs. 1e, f).

To corroborate the possibly causative role of S100a8/a9 deficiency for enteric infectious susceptibility we studied WN *S100a9*^-/- pups and found them similarly susceptible to severe enteric infections with either *C. rodentium* (Supplementary Figs. 4a–d) or *S. typhimurium* (Supplementary Fig. 4e-h) as MN wild type mice (Fig. 2). Like in the comparison of MN to WN wild type pups, WN *S100a9*^-/- pups responded to both of these infections with more severe intestinal inflammation (Supplementary Figs. 4a, c, e, g), stronger LI length restrictions (Supplementary Figs. 4b, f), and higher bacterial translocations into organs (Supplementary Figs. 4d, h) than WN wild type pups. Direct antimicrobial effects of S100a8/a9 against *C. rodentium* and *S. typhimurium* could be excluded (Supplementary Fig. 4i), supporting that S100a8/a9 plays an important immunoregulatory role that protects against the development of enteropathy and susceptibility to enteric infections under maternal malnutrition.

## Nutritional supply of S100a8 after birth prevents the aberrant development of gut immunity and dysbiosis under maternal malnutrition

To test whether S100a8/a9-priming of the neonatal gut can counteract the strong imprinting effect of maternal malnutrition and prevent enteropathy development, neonates from MN dams were fed a single dose of S100a8 within the first 24 h after birth (Fig. 4a). S100a8 homodimers were chosen as they are the most immunoactive form of S100-alarmins[31,43,44] and have no antimicrobial effects (Supplementary Fig. 4i and[42,45,46]). The intestinal expression of Tlr4, the major receptor of S100a8/a9[43,47–49], was comparable in the WN and MN offspring (Supplementary Fig. 5a). Supplemented S100a8 could be retrieved in the feces of MN mice for up to 3 days after feeding (Supplementary Fig. 5b, c), consistent with the extracellular half-life of S100a8/a9 of approximately 24 hours[50,51]. S100a8 treatment significantly reduced their fecal Lcn-2 levels (Fig. 4b) and suppressed the influx of PMNs at d7, particularly in the LI (Fig. 4c), as well as the expansion of proinflammatory Cx3cr1^low LPMPs at d23 in both the SI and LI (Fig. 4d). Contrary, the expansion of regulatory Cx3cr1^high LPMPs in the SI and LI at d23 was significantly improved by the neonatal S100a8 supply (Fig. 4e) as well as their expression of MHC-II in the LI (Fig. 4f). The induction of a less inflammatory but more regulatory state was also reflected by contained *Tnf* (in the SI and LI) but enhanced *Il10* (in the SI significantly, in the LI slightly) and *Tgfb1* (in the SI significantly, in the LI slightly) expression following S100a8-treatment (Supplementary Fig. 6a). Presumably by enhancing the expression of Cxrcr1, Il-10 and

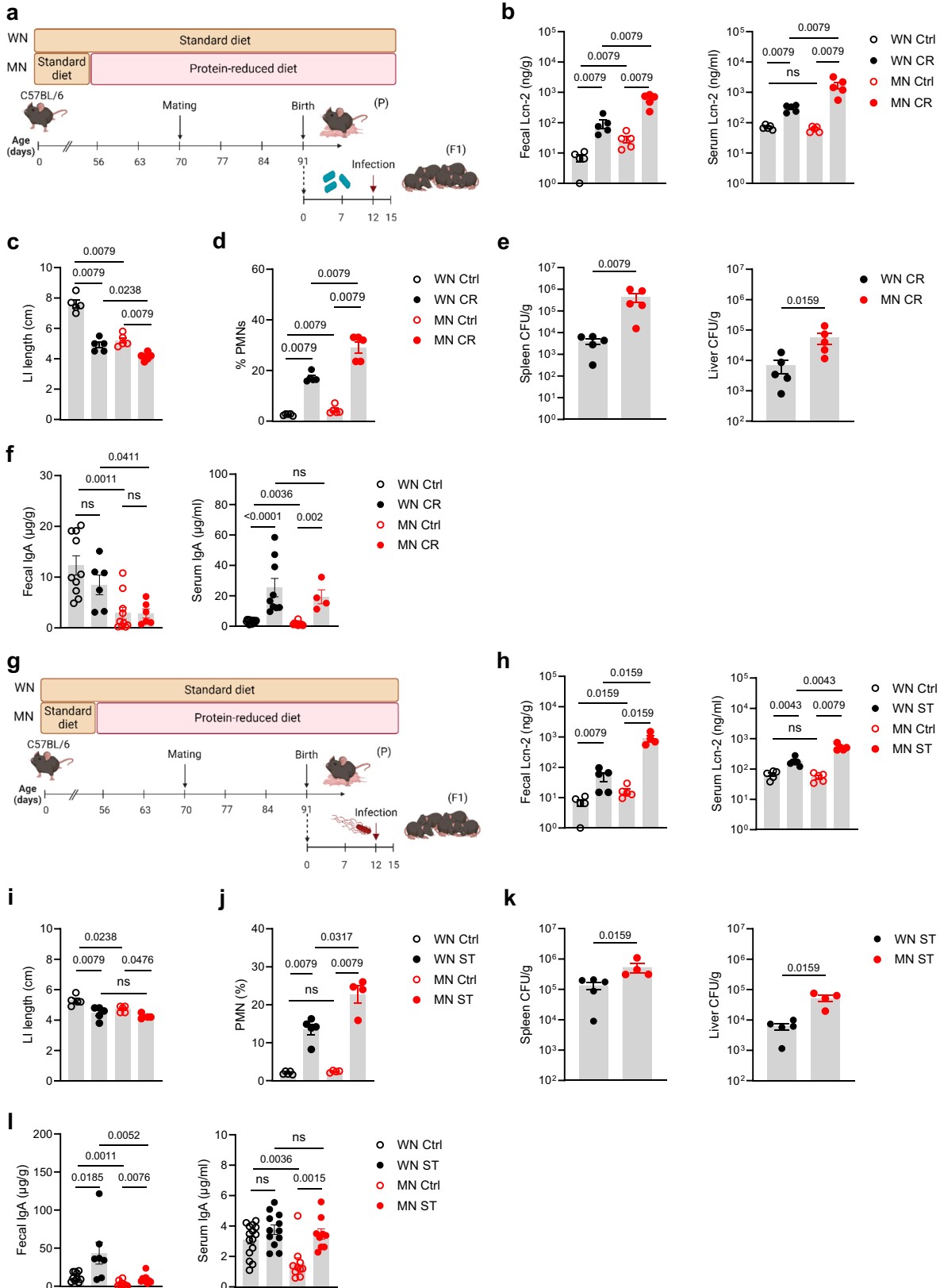

Tgf-β, S100a8 treatment also improved significantly the expansion of intestinal Tregs in the LI of pups of MN mothers, while in the SI a trend was observed (Fig. 4g).

Likewise, a single supply of S100a8 to neonates from MN dams promoted the development of a more favorable gut microbiota state. The amounts of total aerobic and anaerobic bacteria in colon content samples remained unaffected, however, the premature intestinal overgrowth of *Enterobacteriaceae* in the first week of life in MN mice was completely abrogated following a postnatal S100a8 supplementation (Fig. 4h). LDA scores based on 16S rRNA gene sequencing data established from d23 cecal contents revealed an enrichment of certain members of the class *Clostridia*, such as

**Fig. 2 | The early-life enteropathy established under maternal malnutrition aggravates subsequent enteric infections. a** Experimental setup of the *C. rodentium* infection model. **b–f** The offsprings of the MN and WN group were infected at d12 after birth with *C. rodentium* (CR) or treated with PBS (control, Ctrl). Biosamples were harvested 10 days *post infection* (p.i.). **g** Experimental setup of the *S. typhimurium* infection model. **h–l** The offsprings of the MN and WN group were infected at d12 after birth with *S. typhimurium* (ST) or treated with PBS (Ctrl). Biosamples were harvested 3 days p.i. **b, h** Fecal and serum Lcn-2 levels, respectively (*n* = 5 each group). **c, i** LI length (*n* = 5 each group). **d, j** Proportions of PMNs from LI LMPCs (*n* = 5 each group). **e, k** Bacterial load in spleens and livers of infected mice plotted as colony-forming units (CFU) per organ weight (*n* = 5 each group). **f, l** Fecal (Ctrl groups: *n* = 10, infected groups MN: *n* = 6-9) and serum total IgA levels (Ctrl WN: *n* = 15, Ctrl MN: *n* = 10; infected groups: WN: *n* = 9, MN: *n* = 4 (CR) or *n* = 9 (ST)), respectively. Plots represent means ± SEM. Exact *p*-values are displayed, ns, not significant (two-tailed MWU-tests). Panel a and g were created in BioRender under license number BioRender.com/z09x085.

*Lachnospiraceae*, alongside with an increase of amplicon sequence variants (ASVs) within the phylum candidatus *Saccharibacteria* (TM7), the order *Bacteroidales*, and the family *Bacteroidaceae* in the S100a8 treatment group compared to the control group (Fig. 4i). Microbiota abundances most dominantly contained following neonatal S100a8 supplementation included ASVs ascribed to γ-*Proteobacteria*, namely the family *Enterobacteriaceae*, alongside with few members of the orders *Lactobacillales* and *Clostridiales*. Direct comparison of relative abundances at the bacterial family level confirmed a significant reduction of *Enterobacteriaceae* in S100a8-supplemented MN mice, while no S100a8 effect was detectable for *Clostridiaceae* (Fig. 4j).

Finally, a global positive effect on growth and thriving (Supplementary Figs. 6b–d) and intestinal barrier functions (Supplementary Figs. 6e, f) was observed, corroborating the improved gut immune and microbiota state of MN neonates following a single S100a8 feeding.

Overall, these findings show that a single nutritional supplementation of S100a8 to neonates raised by MN dams is sufficient to abrogate the development of severe enteropathy and major microbiota alterations by restoring a more mature intestinal immune phenotype and thus overall gut homeostasis.

### Nutritional S100a8 supply protects neonates fostered by MN mothers against severe enteric infections

To address to the question whether a neonatal S100a8 supplementation also protects a MN offspring against severe enteric infections, neonates from MN mothers were fed once after birth with S100a8 or left untreated and infected at d12 with either *C. rodentium* (Fig. 5a) or *S. typhimurium* (Fig. 5f). In both infection scenarios, S100a8-supplemented MN mice showed less intestinal and also less systemic inflammation as measured by restricted increases of Lcn-2 levels in the stool and blood upon infection compared to untreated MN pups (Figs. 5b, g). Likewise, in both infection settings, LI shortening (Figs. 5c, h) as well as infection-induced influxes of PMNs into the LI mucosa (Figs. 5d, i) were significantly reduced following S100a8 supplementation. This also improved the preservation of intestinal barrier functions during infections as reflected by significantly less bacterial organ dissemination of *C. rodentium* (Fig. 5e) and *S. typhimurium* (Fig. 5j) in S100a8-primed MN mice compared to mock-treated MN mice.

[49]Taken together, under malnutrition conditions, the increased risk of the offspring to suffer from severe enteric infections can be minimized by a single nutritional supplementation of S100a8 directly after birth.

### Maternal malnutrition establishes lifelong colitis susceptibility that can be defeated by a neonatal S100a8 supply

Given the strong imprinting effect of environmental cues including dietary conditions[52–54] on the developing neonatal immune system, we interrogated our model whether maternal malnutrition renders the offspring lifelong susceptible to intestinal insults and whether this is preventable by a neonatal nutritional supplementation of S100a8. Therefore, we followed mice from WN and MN dams either neonatally supplemented with S100a8 or mock-treated until adulthood (Fig. 6a) and assessed the overall and intestinal health status. Mice of all groups and genders had a similar increase of body weights over time after weaning (Figs. 6b, c). However, mice of MN mothers could not make up the impaired gain of body weight experienced during nursing after weaning onto standard diet, resulting in lower absolute body weights at the age of 12 weeks than those of WN mothers (Fig. 6d). Interestingly, at the age of 12 weeks, former MN mice had still significantly higher fecal and even also higher systemic Lcn-2 levels than adult WN mice (Fig. 6e). At the same time, PMNs were expanded in the LI of former MN mice compared to WN mice (Fig. 6f), suggesting that maternal malnutrition causes persistence of a clinically silent inflammatory gut state.

S100a8 supply to neonates from MN dams slightly restored the total body weight gain during adolescence of male but not female mice (Fig. 6d). However, neonatal S100a8 treatment clearly averted the persistence of gut inflammation as reflected by normalized fecal and systemic Lcn-2 levels (Fig. 6e) and restricted abundance of intestinal PMNs (Fig. 6f) in adult former MN mice.

To induce a strong inflammatory insult in these mice at the age of 12 weeks, we employed a DSS colitis model (Fig. 6g). Maternal malnutrition during nursing increased the risk of fatal colitis in adulthood dramatically compared to mice raised by WN mothers; however, former MN mice could be rescued when neonatally supplemented with S100a8 (Fig. 6h). Moreover, colitis-induced LI shortening was strongest in former MN mice, which could also be significantly mitigated by a neonatal S100a8 treatment (Fig. 6i).

Summarized, our findings demonstrate that maternal malnutrition induces a lifelong, clinically silent inflammatory gut state that predisposes long-term to fatal courses of intestinal inflammatory diseases. Nutritional supplementation of S100a8 after birth prevents maternal malnutrition-associated chronic intestinal inflammation, thus conferring long-term protection against later intestinal inflammatory exacerbations in the offspring of MN mothers.

## Discussion

A key complication of malnutrition is the development of EE, which is supposed to be triggered primarily by environmental factors[5,9]. As EE usually becomes clinically overt only after weaning[27], a popular opinion is that children are protected as long as they are breastfed. Improving hygienic conditions failed to prevent EE development[20–22], challenging the concept of frequent uptake of pathogens triggering EE. This is supported by a recent study in rural Kenya demonstrating that intestinal bacterial overgrowth or gut microbiota diversity was not linked to EE[27]. Performing longitudinal immunological and microbiological studies in a murine model of maternal malnutrition, this study revealed that a chronic inflammatory enteropathy is established already during nursing in neonates fostered by MN mothers and paves the way for severe enteric infections and aberrant gut colonization.

Our data demonstrated that such maternal malnutrition is sufficient to induce in the offspring growth retardation, intestinal barrier dysfunction and a persistent inflammatory gut mucosal phenotype with an expansion of proinflammatory Cx3cr1[low] LPMPs but lack of regulatory Cx3cr1[hi] LPMPs and Tregs. In contrast, in models applying malnutrition only in weaned mice, additional microbial challenges were necessary to induce an inflammatory enteropathy[18,19]. Interestingly, malnutrition models restricted to

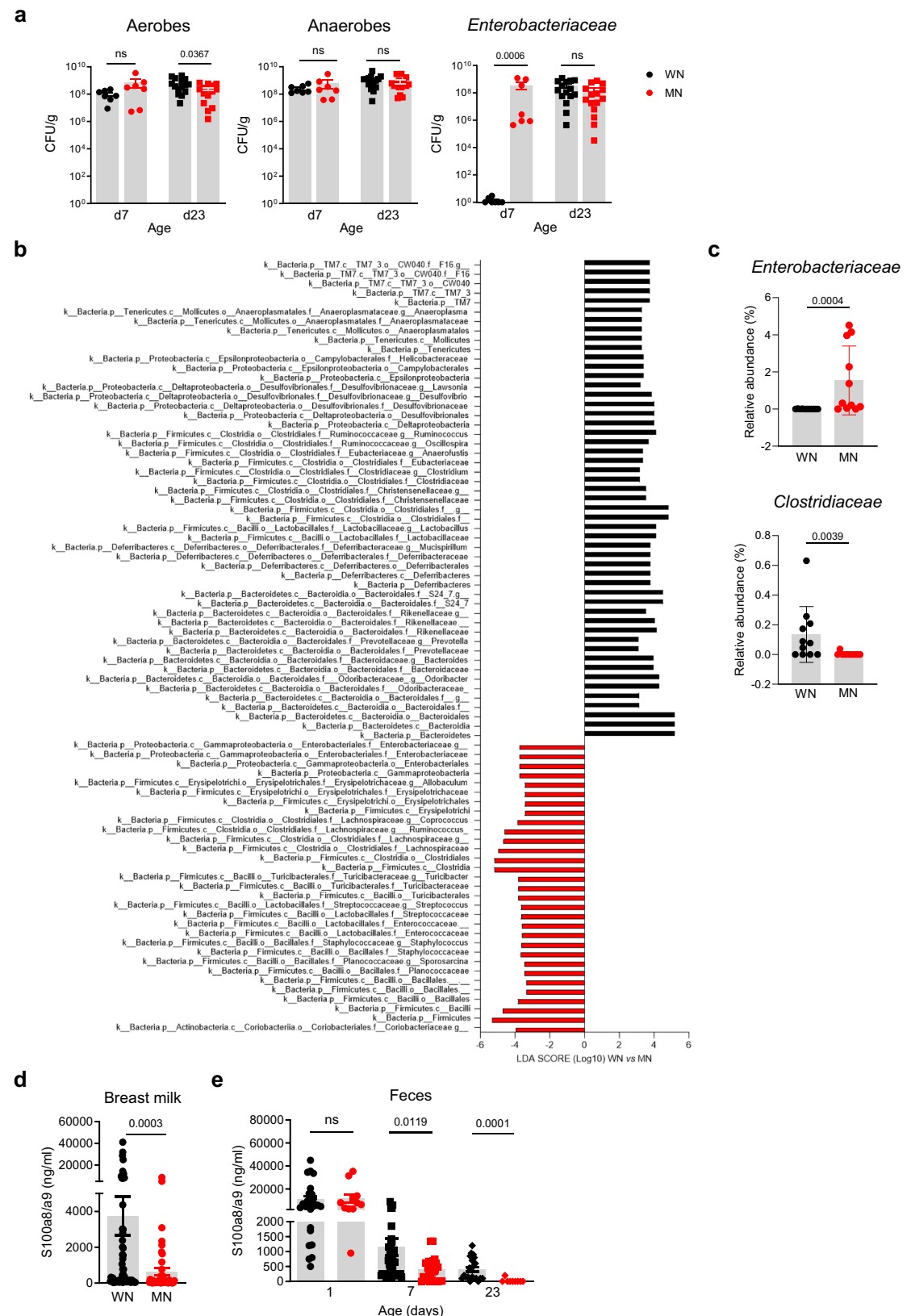

**a**

Aerobes, Anaerobes, *Enterobacteriaceae* (CFU/g vs Age d7, d23); WN (black), MN (red)

**b** LDA SCORE (Log10) WN *vs* MN

**c** *Enterobacteriaceae* (Relative abundance %, WN vs MN, 0.0004); *Clostridiaceae* (Relative abundance %, WN vs MN, 0.0039)

**d** Breast milk (S100a8/a9 (ng/ml), WN vs MN, 0.0003)

**e** Feces (S100a8/a9 (ng/ml) vs Age (days) 1, 7, 23; ns, 0.0119, 0.0001)

pregnancy allowed catch-up growth in the offspring fostered by WN mothers[55], while this was not the case in our model when mice fostered by MN mothers were weaned onto standard diet. Given this background, our study highlights the postnatal period as the most vulnerable phase of life for the establishment of gut homeostasis under malnutrition conditions.

The causes for the heightened risk of severe enteric infections in MN children[5–7] are ill defined. Malnutrition-mediated deterioration of immune functions in the sense of 'immunodeficiency', especially in the T cell compartment, have been discussed yet not shown in the context of enteric infections[8,56]. Regarding the concept of dysbiosis as precursor to infection, trials using antimicrobial prophylaxis or hygiene

**Fig. 3 | Development of gut dysbiosis and S100a8/a9 deficiency under maternal malnutrition. a** Abundance of total bacteria cultivated under aerobic and anaerobic conditions and of *Enterobacteriaceae* in fecal samples (LI plus cecum contents) collected from the offspring of WN and MN dams at indicated time points (d7: $n = 7$ each group, d23: $n = 15$ each group). Bars represent means ± SEM. Exact *p*-values are displayed, *ns*, not significant (two-tailed MWU-tests). CFU, colony-forming units. **b** Linear discriminant analysis (LDA) effect size (LEfSe) in the WN *versus* MN group (each $n = 11$) on relative abundances of gut microbiota in cecum contents collected at d23. Plotted are LDA scores for discriminative features >3.6 at phylum, class, family and genus levels. **c** Relative abundance of *Enterobacteriaceae* and *Clostridiaceae* in d23 cecum contents of the WN and MN offspring ($n = 11$ each group). **d** S100a8/a9 levels in breast milk of WN and MN dams collected during d1-d7 after delivery (WN: $n = 63$, MN: $n = 49$). **e** S100a8/a9 levels in the offspring's feces at d1 (WN: $n = 26$, MN: $n = 11$), d7 (WN: $n = 43$, MN: $n = 13$) and d23 (WN: $n = 21$, MN: $n = 12$). Bars represent means ± SEM. Exact *p*-values are displayed, *ns*, not significant (two-tailed MWU-tests).

interventions could not demonstrate public health benefit so far[57]. Our data suggested that the underlying proinflammatory gut mucosal state induced by maternal malnutrition predisposes the offspring in an additive, exacerbating manner to severe disease courses, whereas infectibility and infection-induced inflammatory net responses are comparable.

Furthermore, our data do not support the concept of unfavorable gut microbiota colonization triggering EE but rather pointed to a mutual co-evolution of both an inflammatory enteropathy and gut dysbiosis as a result of maternal malnutrition. Hallmarks of gut dysbiosis in our model were premature expansion of pathobiontic *Enterobacteriaceae* at the cost of commensal *Clostridiaceae*. The relevance is corroborated by human studies reporting increased abundances especially of *Enterobacteriaceae* in MN children[15,17]. Such unfavorable gut microbiota state might certainly promote and maintain the inflammatory gut phenotype[58], however, assigning gut dysbiosis a truly causative role in EE development needs further clarification in follow-up studies.

Both the cellular gut mucosal alterations as well as the premature expansion of *Enterobacteriaceae* found in neonates fostered by MN mothers we previously identified as key features in neonates fostered by S100a8/a9-deficient WN mothers[29]. Here, we provided for the first time evidence that neonates from MN mothers undergo deficient intestinal S100a8/a9-priming due to reduced S100a8/a9 levels in the breast milk. This disproves hitherto studies claiming that maternal undernutrition has little effect on the composition of breast milk[2,26] and questions recommendations of exclusive breastfeeding for the first 6 months in developing countries[59]. We showed that a single feeding of S100a8 to neonates fostered by MN mothers prevents the development of an enteropathy and related susceptibility to severe enteric infections. Of note, while averting the early overgrowth of *Enterobacteriaceae*, S100a8 supplementation could not promote the expansion of *Clostridiaceae*, suggesting that additional factors are lacking to fully restore a WN intestinal phenotype in an MN offspring.

S100A8/A9 binds and activates TLR4 signaling[43,47–49]. In neonates, continuous S100A8/A9-TLR4-signaling tolerizes blood monocytes[31] and induces a regulatory phenotype in intestinal LPMPs, which promotes the expansion of Tregs and is associated with a controlled expansion of *Gammaproteobacteria* like *Escherichia coli*[29]. The latter overgrows likewise in the gut of TLR4 knockout mice[60]. Continuous TLR4 signaling is also known to regulate the proliferation and differentiation of the intestinal epithelium[61,62] and limit bacterial translocation, e.g., through restricting the formation of Goblet cell-associated antigen passages[63–65]. Continuous S100A8/A9-TLR4 signaling in the neonatal gut might therefore also protect from dysbiosis and bacterial translocation by impacting on the intestinal epithelium which future studies must validate experimentally.

The imprinting impact of maternal malnutrition was underlined by demonstrating persistence of clinically silent intestinal inflammation after weaning, which led to a lifelong increased susceptibility to colitis. A nutritional supply of S100a8 after birth protected the offspring of MN mothers long-term from severe colitis, underpinning the manifestation of disease susceptibilities early in life and the opportunity of early intervention when causative factors have been identified.

Since S100a8/a9 has an important regulatory role in both murine and human neonates[29,31,42–44], our findings encourage to consider nutritional S100A8 supplementation to human newborns raised under malnutrition conditions as a simple, cost-efficient treatment to protect them against the chronic inflammatory enteropathy associated with maternal malnutrition and related lifelong health risks.

Protein deficiency was the main dietary alteration caused in MN mothers. Albeit to a lesser extent, the diet applied for malnutrition in our model was also deficient in fat and enriched in carbohydrates and additionally purified in contrast to the grain-based standard diet, which collectively might have contributed to the immunological and microbial alterations observed in the offspring of MN mothers[66]. In this context, particularly reduced supply of fatty acids and vitamin A might have contributed to the alteration in intestinal structures and gut flora of the offspring[67,68]. However, studies that have focused on maternal fatty acid or vitamin A deficiencies differ from our results with respect to the microbiota alterations in the offspring. For example, contrary to what we observed in our model in the offspring of MN dams, maternal deficiency of omega-3 long-chain polyunsaturated fatty acids has been associated with a higher relative abundance of *Clostridiaceae*[68] and vitamin A deficiency neither altered the abundance of *Enterobacteriaceae* nor *Clostridiaceae*[67]. To clarify to what extent biologically active proteins other than S100A8/A9 or lipid species or vitamins influence the gut phenotype developing under malnutrition follow-up studies with more refined dietary alterations of the mothers are required.

In summary, our data demonstrated that maternal malnutrition with a strong reduction in protein supply induces a S100a8/a9 deficient state in neonates that promotes the development of a persistent chronic inflammatory enteropathy that including associated health risks can be prevented by a single nutritional S100a8 supplementation after birth. Our findings encourage to perform clinical trials testing the safety and efficacy of nutritional S100A8 supplementation to improve the overall health outcome of children born under MN conditions.

## Methods

### Mice and mouse model of maternal malnutrition

For all experiments employing the model of maternal malnutrition, C57BL/6 mice (Charles River, Sulzfeld, Germany or Charles River, Calco, Italy) or a *S100a9*$^{-/-}$ knock-out stain (B6.S100a9$^{tmINck}$, provided by T.V.) were used. Until mating, mice were housed individually under controlled, specific pathogen free conditions (SPF) in barrier protected rooms with a 14 h light and 10 h dark cycle at two different animal laboratories, at the Hannover Medical School (Germany) and Institute for Research in Biomedicine (Switzerland). The room temperature was kept at $22 ± 2$ °C with an air humidity of $55 ± 5\%$. Before start and after completion of the experiments, mice were fed *ad libitum* with a pelleted standard animal chow. Sterile drinking water was offered in drinking bottles. To model maternal malnutrition, we established parental malnutrition, in which adult male and female mice (aged >8 weeks) were fed a primarily protein-reduced but also fat-reduced and carbohydrate-enriched diet (S7088-E710, ssniff, Germany, Supplementary Table 1) *ad libitum*, starting 14 days prior to mating and kept on it until weaning of the litters (MN group). The WN control group was fed standard chow (1314, Altromin, Germany,

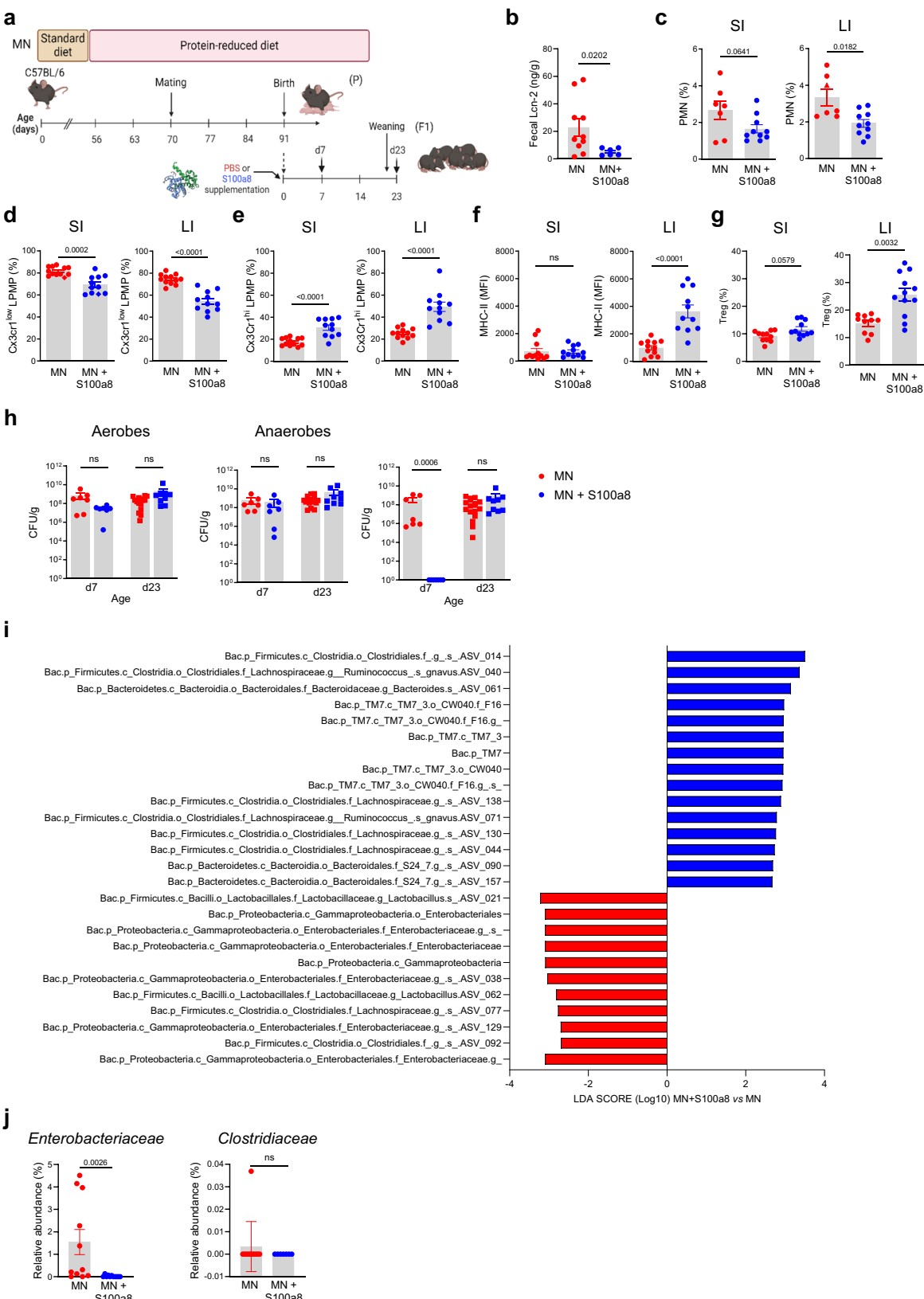

WN respective MN dams. Both sexes were used equally and subsetted where appropriate.

**Breast milk collection.** In dams of both MN and WN groups, breast milk was collected over day 1–7 after delivery by gently massaging the lower teats and afterwards centrifuged to separate the milk fat. The

Supplementary Table 1) and mated in parallel to the MN group. Chow consumption was monitored every second to third day by weighing remaining chow. Body weights of parental mice and the offspring were documented every second to third day. The offspring's tail length was measured at the same time points as surrogate of growth. The offspring was named WN respective MN pups or mice when fostered by

**Fig. 4 | Nutritional S100a8 supply after birth promotes healthy development of intestinal immunity and gut microbiome under maternal malnutrition.**
**a** Experimental setup. Pups of MN dams were fed 5 µg of S100a8 within 24 h after birth (S100a8) or control treated with PBS (Ctrl). **b** Lcn-2 levels in fecal samples at d23 (MN: *n* = 10 mice, MN+S100a8: *n* = 6 mice). **c**–**g** Flow cytometric analysis of LPMCs isolated from the SI and LI at d7 (MN: *n* = 7, MN+S100a8: *n* = 10) and d23 (MN: *n* = 12, MN+S100a8: *n* = 11). **c**–**e** Proportions of PMNs at d7 from LPMCs (**c**) and Cx3cr1^low LPMPs (**d**) and Cx3cr1^hi LPMPs (**e**) at d23 from LPMPs. **f** Expression of MHC-II on Cx3cr1^hi LPMPs (MFI). **g** Proportions of Tregs from LPMCs at d23. **h** Abundance of total bacteria cultivated under aerobic and anaerobic conditions

and of *Enterobacteriaceae* in fecal samples (LI plus cecum contents) collected at indicated time points from S100a8-treated and PBS-treated MN mice (d7: *n* = 7 each group, d23: MN: *n* = 15, MN+S100a8: *n* = 9). CFU, colony forming units. **i** LDA scores for relative bacterial abundances in d23 cecum contents with discriminative features >3.6 at phylum, class, family and genus levels between S100a8-supplemented *versus* not supplemented MN mice (*n* = 7-11 each group). **j** Relative abundance of *Enterobacteriaceae* and *Clostridiaceae* in d23 cecum content (*n* = 11 mice per group). Bars represent means ± SEM. Exact *p*-values are displayed, *ns*, not significant (two-tailed MWU-tests). Panel (**a**) was created in BioRender under license number BioRender.com/p20q301.

liquid phase was then stored at -80 °C until quantification of S100a8/a9.

### S100a8 supplementation.
In indicated experiments, newborn pups from MN dams were supplemented within 24 h after birth by oral feeding of 5 µg of recombinant murine S100a8 in 20 µl PBS. Control pups were supplemented with 20 µl PBS.

### Collection of biosamples in the offspring.
At the age of day 1, day 7 and day 23, serum, intestine, and feces were harvested from the offspring. The intestine was separated into SI and LI. Length of both SI and LI were measured using a standard ruler (0.1 cm interval). Intestinal content was carefully removed from the SI, cecum, and LI using tweezers and aliquots stored at −80 °C until further analysis.

### Isolation of LPMCs.
Isolation of LPMCs was performed as described previously[29]. Briefly, intestinal tissues were washed once in medium (RPMI 1640 with 10% FCS and 1% penicillin/streptomycin) and then minced into small pieces. Epithelia were removed by using PBS with 2 mM EDTA for 15 min. Afterwards, intestine pieces were digested using digestion medium (RPMI 1640 (Sigma Aldrich) with 50 µg/ml Liberase™ (Sigma Aldrich) and 30 U/ml DNAse I (Millipore)) for 10 min (SI) or 50 min (LI). Isolated single-cell suspensions were used for FACS analyzes or preparation of RNA lysates for qRT-PCR.

### Mice and mouse models of enteric infections
C57BL/6 mice of the WN and MN group and in indicated experiments WN wild type C57BL/6 pups and WN *S100a9*^−/− pups (C57BL/6 background)[69] were randomly assigned to the not infected (control) or infected experimental group. In the *C. rodentium* model, 12 days old mice were orally infected with $10^8$ colony-forming units (CFU)/mouse of *C. rodentium* or treated with PBS. After 10 days from the beginning of the experiment, the mice were sacrificed by inhalation of $CO_2$ for the retrieval of organs (intestines, spleens, livers), blood and fecal samples. In the *S. typhimurium* model, 12 days old mice were orally infected with $10^6$ CFU/mouse of *S. typhimurium* or treated with PBS. After 3 days from the beginning of the experiment, the mice were sacrificed by inhalation of $CO_2$ for the retrieval of organs (intestines, spleens, livers), blood and fecal samples.

For determination of total *C. rodentium* CFU loads, homogenates of spleen and liver, collected at 10 days post infection, were plated on LB agar plates containing 12.5 mg/ml of tetracycline and placed at 37 °C for 24 h before enumeration of colonies. For determination of total *S. typhimurium* CFU loads, homogenates of spleen and liver, collected at 3 days post infection, were plated on MacConkey agar plates containing 50 µg/ml streptomycin and placed at 37 °C for 24 h before enumeration of colonies.

### DSS-induced colitis mouse model
Mice of the WN and MN group were weaned at d21 onto standard diet and observed for further 9 weeks. At the age of d84, DSS (TdB LabsAB, Uppsala, Sweden) was added at a concentration of 2.5 % to the drinking water for 6 days. Then, DSS 2.5% was replaced by normal drinking

water. Survival was monitored for 14 days after start of DSS treatment. Surviving mice were sacrificed by inhalation of $CO_2$ for the retrieval of organs.

### Ethics statement
Mouse experiments were performed according to the Swiss Federal Veterinary Office guidelines and the German Animal Welfare Legislation and has been approved by the Cantonal Veterinary Office (approval no. 31975 TI-19/2020/2023; 34029 TI-40/2021; 34591 TI-02/2022) and the Lower Saxony State Office for Consumer and Food Safety, Germany (approval no. 20/3366; 21/3683).

### Bacterial cultivation
The tetracycline-resistant HA538 strain (Ddadx::tetRA) of *C. rodentium* was cultured on LB agar plates supplemented with tetracycline (12.5 mg/ml) and then expanded in Luria broth overnight at 37 °C under continuous agitation. *S. typhimurium* SL1344 (clone SB300) was cultured on LB agar plates supplemented with streptomycin (Sigma Aldrich) at 50 mg/ml and then expanded in Luria broth overnight at 37 °C under continuous agitation.

### Bacterial growth inhibition assay
In order to test direct antimicrobial activity, S100a8/a9 and S100a8 were diluted in HBSS (without $Ca^{2+}$ and $Mg^{2+}$) at 100 µg/ml while bacterial suspensions of *C. rodentium* HA538 and *S. typhimurium* SL1344 were diluted ten-fold three times in RPMI 1640 medium with 25 mM HEPES. Equal volumes of both preparations were mixed and 200 µL culture aliquots grown at 37 °C while shaking at 180 rpm in a 96-well microtiter plate. Bacterial suspensions were mixed with HBSS supplemented with 5 µg/ml S100a8 or 50 µg/ml S100a8/a9 50 or HBSS without S100a8 or S100a8/a9 (controls). Aliquots of the cultures were taken after 0, 2, 5, 8 and 24 h, diluted and plated onto agar plates to determine the number of CFU, respectively. Additionally, bacterial growth was monitored by measuring the increase in optical density at 600 nm ($OD_{600}$) over time.

### Quantification of Lcn-2, IgA and S100a8/a9
For measurement of lipocalin-2 (Lcn-2) and IgA per ELISA assays, serum samples were used without further processing. Fresh fecal samples were suspended in PBS (0.1 g/ml) and centrifuged twice for 10 min at maximum speed, and the supernatant was stored at −20 °C till use. The ELISA assay for Lcn-2 was purchased from R&D systems (DuoSet ELISA Mouse Lipocalin-2/NGAL). Serial dilutions of all samples were used to perform the Lcn-2 ELISA assay following the manufacturer's specifications. For IgA quantification, ELISA plates (3690, half-area 96 well plate, Corning, USA) were coated (16 h at 4 °C) with 25 ml of unlabeled goat anti-murine IgA (Cat: 1040-01, Southern Biotech) at 5 mg/ml in PBS, washed 4 times with PBS 0.025% Tween 20 (Cat: 93773, Sigma Aldrich) and saturated with 50 mL of PBS 1% BSA for 1 h at room temperature. Twenty-five ml of serial dilutions of the different samples were incubated 2 h at room temperature. After 4 washes in PBS 0.025% Tween 20, 25 ml of alkaline phosphatase (AP) conjugated goat

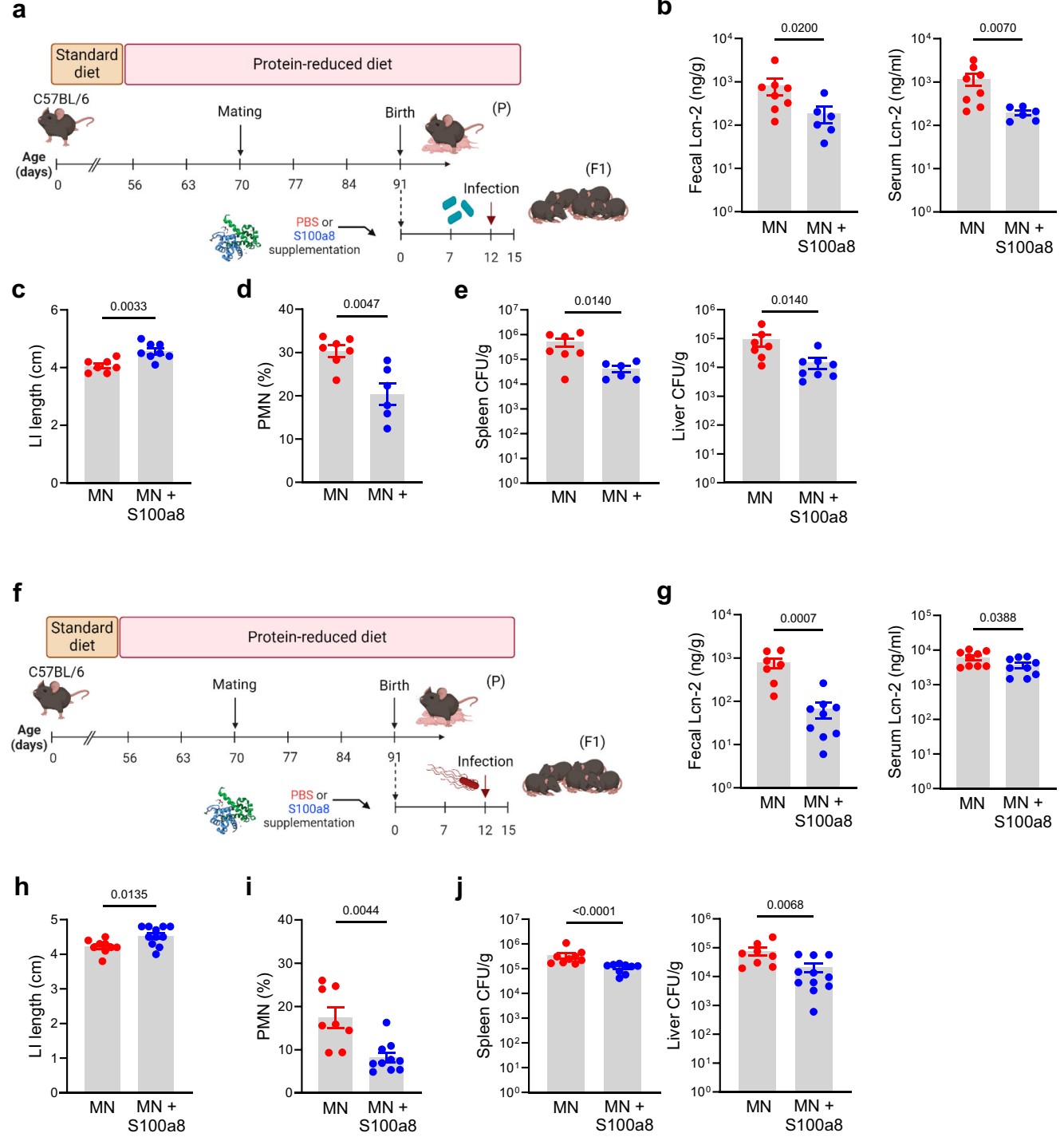

**Fig. 5 | S100a8 supplementation after birth lowers the risk of severe early-life enteric infections under malnutrition conditions. a** Experimental setup of the *C. rodentium* infection model with S100a8 supplementation. **b**–**e** Neonatally S100a8-treated and untreated mice from MN dams were infected at d12 with *C. rodentium* (CR) or treated with PBS (Ctrl). Biosamples were harvested 10 days p.i. (MN: *n* = 8, MN+S100a8: *n* = 6-8). **f** Experimental setup of the *S. typhimurium* infection model with S100a8 supplementation. **g**–**j** Neonatally S100a8- treated and untreated mice from MN dams were infected with *S. typhimurium* at d12 (ST) or treated with PBS (Ctrl). Biosamples were harvested 3 days p.i. (MN: *n* = 8, MN +S100a8: *n* = 11). **b**, **g** Fecal and serum Lcn-2-levels, respectively. **c**, **h** LI length. **d**, **i** Proportions of PMNs from LI LMPCs. **e**, **j** Bacterial load in spleens and livers of infected mice plotted as CFU per organ weight. Plots represent means ± SEM. Exact *p*-values are displayed, *ns*, not significant (two-tailed MWU-tests). Panel a and f were created in BioRender under license number BioRender.com/i12p840.

anti-mouse IgA (Cat: 1040-04, Southern Biotech) 1:500 in PBS 1% BSA was added and plates were incubated for 2 h at room temperature. The assay was developed with 4-Nitrophenyl Phosphate disodium salt hexahydrate (Cat: N2765-100TAB, Sigma Aldrich) in carbonate buffer, composed by 1.59 g Na2CO3/l deionized water (Cat: 1.06395.0500, Merk, Germany) and 2.93 g NaHCO3/l

deionized water (Cat: A0384,0500, AppliChem, Germany), and absorbance was detected at 405 nm.

S100a8/a9 was quantified in breast milk, fecal and blood plasma samples using an in-house ELISA described previously[43,48]. Harvested blood plasma and breast milk samples were used in the ELISA assay without further processing. Fecal samples were suspended in

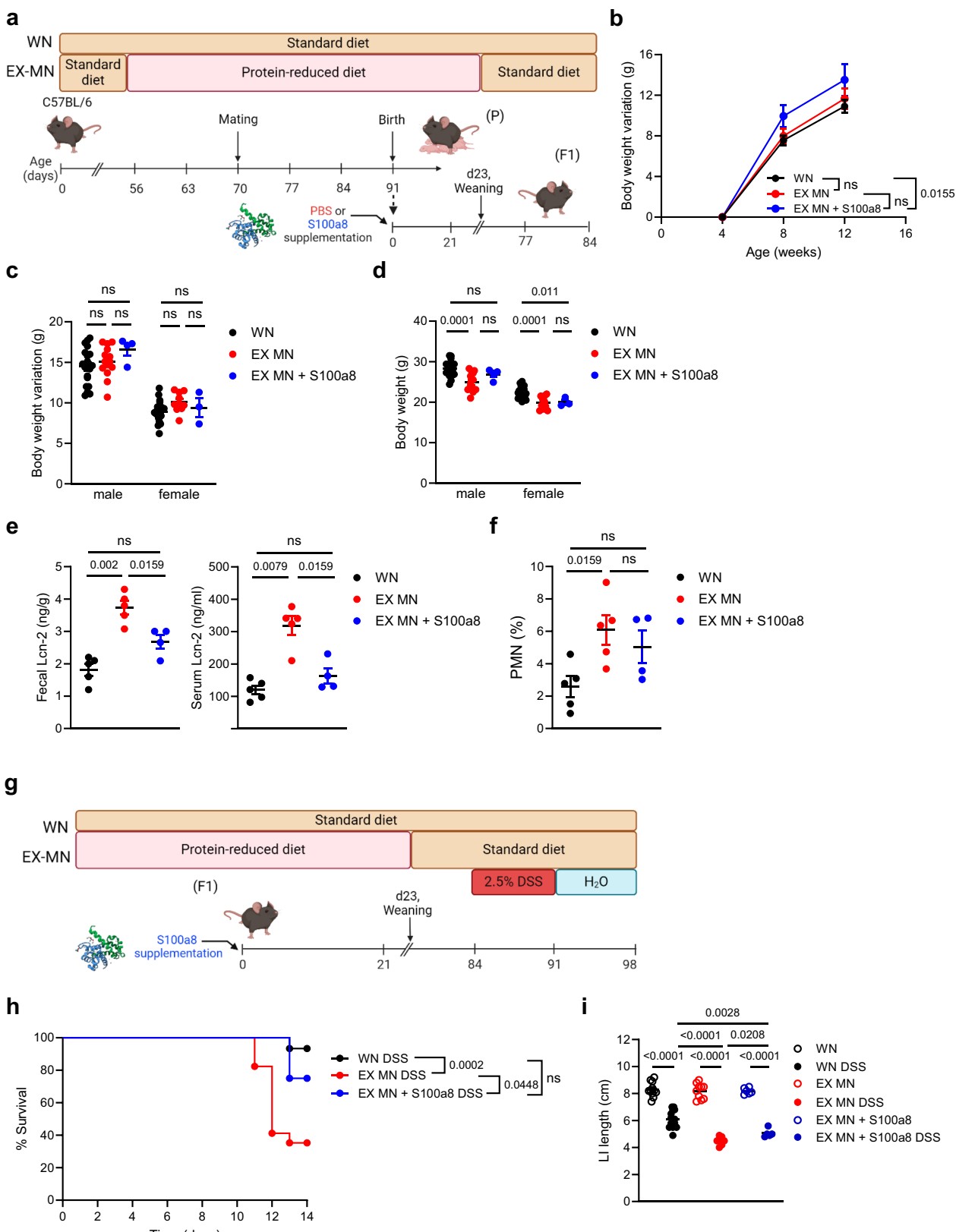

extraction buffer adopted from Hycult Biotech's H325 Human Cal-protectin ELISA kit. Suspensions were thoroughly vortexed, filtered through a 100 μm cell strainer, and then incubated on a shaker on ice for 20 min. The homogenates were centrifuged for 20 min at 10,000 × g at 4 °C. The clear part of the supernatant was pipetted off and stored at −80 °C until measurement.

## Immunoblotting

Homogenates of fecal samples (each 10 μl) were subjected to SDS-PAGE and Western blot staining as described earlier[31,43] using a primary polyclonal anti-murine S100a8 antibody purified by T.V. The anti-rabbit horseradish peroxidase conjugated secondary antibody was obtained from Cell Signaling (Leiden, Netherlands). Protein bands

**Fig. 6 | Lifelong colitis susceptibility is a sequela of maternal malnutrition and preventable by supplementing S100a8 after birth. a** Experimental setup of the follow-up of the group of WN mice, not supplemented former MN mice (EX-MN) and postnatally S100a8-supplemented former MN mice (EX-MN + S100a8). **b–f** WN mice, EX-MN and EX-MN + S100a8 were all weaned onto standard diet and monitored till the age of 84 days. **b** Body weight variations over time from 4 weeks until d84 (WN: $n = 29$, EX-MN: $n = 11$ and EX-MN+S100a8: $n = 7$). **c, d** Body weight variations from 4 weeks until d84 (**c**) and absolute body weight at d84 (**d**), respectively separated by gender (WN male: $n = 20$, WN female: $n = 24$, EX-MN male: $n = 14$, EX-MN female: $n = 10$, EX-MN+S100a8 male: $n = 4$ and EX-MN+S100a8 female: $n = 3$). **e** Lcn-2 levels in feces and serum at d84 (WN: $n = 5$, EX-MN: $n = 5$, EX-MN+S100a8: $n = 4$). **f** Proportions of PMNs from LI LPMCs (WN: $n = 5$, EX-MN: $n = 5$, EX-MN

+S100a8: $n = 4$). Plots represent means ± SEM. Exact *p*-values are displayed, *ns*, not significant (one-way ANOVA, p*ost hoc* Tukey's multiple comparison test). **g** Experimental setup in the DSS-induced colitis model. **h–i** Colitis was induced in all experimental groups at d84 by daily enteral treatment with 2.5% DSS for 7 days. Control animals were treated with PBS. Subsequently, all mice were enterally treated with water until d98. **h** Survival of colitis until 14 days after start of DSS treatment (WN: $n = 15$, EX-MN: $n = 17$, EX-MN+S100a8: $n = 8$). Exact *p*-values are displayed, *ns*, not significant (Mantel-Cox test). **i** LI length at d98 (WN Ctrl: $n = 11$, WN DSS: $n = 14$; EX-MN: $n = 10$, EX-MN DSS: $n = 13$, both EX-MN+S100a8 groups: $n = 6$). Plots represent means ± SEM. Exact *p*-values are displayed, *ns*, not significant (one-way ANOVA, *post hoc* Tukey's multiple comparison test). Panel (**a** and **g**) created in BioRender under license number BioRender.com/c14y754.

were visualized using the enhanced chemiluminescence system and the ChemiDoc Imaging System with Image Lab Software v. 6.1 (Bio-Rad Laboratories, Germany)).

### Flow cytometry
FACS analysis was performed using single-cell suspensions of SI and LI LPMC as described previously[29]. In all staining panels, the Fixable Viability Dye eFluor 506 (eBioscience) was used for the exclusion of dead cells. CD16/CD32 (2.4G2, BioLegend) was used for blocking purposes. All stainings were performed for 30 min in the dark at 4 °C. After surface staining, cells were fixed using 2% PFA.

**Polymorphonuclear neutrophils (PMNs).** For the detection of CD45+CD11b+Ly6G+CD11c- PMNs, LPMCs were stained with rat anti-mouse CD45 mAb (30F-11), hamster anti-mouse CD11c mAb (N418), rat anti-mouse CD11b mAb (M1/70) (all eBioscience), and rat anti-mouse Ly6G mAb (1A8, BioLegend), and gated as illustrated in Supplementary Fig. 7a.

**LPMPs.** For the detection of CD45+F4/80+CD11b+/-Ly6G-CD11c- LPMPs, LPMCs were stained with rat anti-mouse CD45 mAb (1:600, 30F-11), rat anti-mouse F4/80 mAb (1:100, BM8), hamster anti-mouse CD11c mAb (1:400, N418), rat anti-mouse CD11b mAb (1:600, M1/70) (all eBioscience), and rat anti-mouse Ly6G mAb (1:200, 1A8, BioLegend), and gated as illustrated in Supplementary Fig. 7b and described previously[29]. Additional staining with mouse anti-mouse Cx3cr1 mAb (1:600, SA011F11, BioLegend) and rat anti-mouse MHC-II mAb (1:600, M5/114.15.2, BD) allowed subsetting LPMPs into proinflammatory Cx3cr1low LPMPs and regulatory Cx3cr1hi LPMPs[33] and assessing expression of MHC-II.

**Tregs.** For the detection of CD3+CD4+FoxP3+ Tregs, LPMCs were first stained extracellularly using rat anti-mouse CD45 mAb (1:600, 30-F11), hamster anti-mouse CD3e mAb (1:200, 145-2C11) and rat anti-mouse CD4 mAb (1:600, RM 4-5) (all eBioscience). For the intracellular staining, cells were fixed in 2% PFA for 15 min at room temperature. Subsequently, cells were stained with rat anti-mouse FoxP3 mAb (1:100, FJK-16s, eBioscience) in Intracellular Staining buffer (FACS buffer with 0.5% saponin and 0.2% Tween20) for another 30 min at 4 °C in the dark and gated as illustrated in Supplementary Fig. 7c.

**Data acquisition and analyzes.** All flow cytometry analyses were performed using a FACS Canto II (BD Biosciences) flow cytometer. Data were analyzed using the FlowJo software or FACS DIVA software v8.0.1 (both BD Biosciences).

### Immunofluorescence microscopy
Formalin-fixed, paraffin-embedded tissue sections of murine LI were deparaffinized using Roti-Histoclear (Roth) and rehydrated. After antigen retrieval using citrate buffer, tissue sections were washed and blocked using 5% skim milk powder in tris-buffered saline (TBS) to prevent nonspecific binding. For immunofluorescence staining, slides

were stained at 4 °C overnight, using rat anti-mouse Ly6G (RB6-8C5, Santa Cruz) and rabbit anti-mouse S100a9 polyclonal antibody (purified by T.V.) or rabbit anti-mouse Tlr4 polyclonal antibody (Novus), followed by AlexaFluor488 goat anti-rat or AlexaFluor555 donkey anti-rabbit (both Invitrogen) secondary antibodies. Tissue was mounted using VECTASHIELD Antifade Mounting Medium with 4′,6-diamidino-2-phenylindole (DAPI) (Vector Laboratories). Images were acquired and analyzed using the Keyence BZ-X800 fluorescence microscope and standard analysis software tool. At least 5 randomly selected images acquired from proximal, mid, and distal LI sections prepared from two d7 mice from two different litters were analyzed per condition. The same exposure times were used for the acquisition of the read-out channels, respectively.

### Quantitative real-time PCR
Total RNA was isolated from murine LPMC lysates using the NucleoSpin RNA II Kit (Macherey-Nagel, Germany) following the manufacturer´s recommendations. cDNA was synthesized from 300 ng of total RNA using the RevertAid Revese Transcriptase Master Mix (ThermoFisher Scientific) following the manufacturer´s recommendations. qRT-PCR was done as described previously[29]. The murine primers used for qRT-PCRs were *Gapdh* (F, GGACACTGAGCAAGA-GAGGC; R, TTATGGGGGTCTGGGATGGA), *Tnf* (F, GATCGGTCCC-CAAAGGGATG; R, GTGGTTTGTGAGTGTGAGGGT), *Il10* (F, GGGTTGCCAAGCCTTATCG; R, TCTCACCCAGGGAATTCAAATG), and *Tgfb1* (F, AGGAGACGGAATACAGGGCT; R, ATGTCATGGATGGTGCC-CAG). Sample data are presented as target gene expression relative to the housekeeper *Gapdh*.

### FITC-dextran assay
Intestinal permeability was assessed using FITC-dextran assay. Mice were starved for 4 h before treatment and then gavaged with 6 mg FITC-dextran (molecular weight 4000; Sigma Aldrich) in PBS per mouse. Blood was collected 4 h post-treatment and the fluorescence intensity was measured in the serum at 485/530 nm using a micro-plate reader (Biotek Synergy 2).

### Microbiological analysis of murine fecal samples
To determine the number of CFUs per gram of intestine content, LI plus cecum content was suspended at 1 ml PBS/g content. Serial dilutions were plated on Columbia agar plates and Schaedler agar plates (both ThermoFisher Scientific) and incubated at 37 °C overnight under aerobic (with 10% CO$_2$) or anaerobic conditions using Anaerocult® C (Merck, Germany) in an airtight jar. For selective growth of *Enterobacteriaceae* diluted samples were plated on MacConkey agar (AppliChem, Germany) and incubated at 37 °C overnight under aerobic (with 10 % CO$_2$) conditions.

### 16S rRNA gene bacterial profiling
Extraction, lysis, and DNA isolation were carried out utilizing the Fast DNA Stool Mini Kit (Qiagen) in strict accordance with the manufacturer's recommendations. Bead beating was performed using a

FastPrep24 instrument (MPBiomedicals), with four cycles of 45 seconds each at a speed of 4. The bead-beating process was conducted in 2 ml screw-cap tubes containing 0.6 g of 0.1 mm glass beads. A total of 200 ml of raw extract was prepared for subsequent DNA isolation.

To assess the concentration of the isolated DNA, we employed PicoGreen measurement, employing the Quant-iTT PicoGreenT dsDNA Assay Kit from Thermo Fisher. Furthermore, to verify the integrity of a representative sample, agarose gel electrophoresis was conducted.

For the amplification of the bacterial 16S rRNA gene, a specific primer set targeting the V3–V4 hypervariable regions was employed. The forward primer (Fw) utilized in this study was 5'-CCT ACG GGN GGC WGC AG-3' (SEQ ID NO: 4), and the reverse primer (Rev) was 5'-GAC TAC HVG GGT ATC TAA TCC-3' (SEQ ID NO: 5). Sequencing of the PCR libraries was performed on the Illumina MiSeq platform using a v2 500 cycles kit.

## Bioinformatics of 16S rRNA gene sequencing data
The generated paired-end reads, which successfully passed Illumina's chastity filter, underwent de-multiplexing and the removal of Illumina adapter remnants. These processes were executed using Illumina's real-time analysis software integrated into the MiSeq reporter software v2.6. No further post-processing or selection was applied at this stage.

To assess the quality of the reads, FastQC version 0.11.8 software was employed. The initial dataset consisted of a total of 363,773 sequences (with a median read count of 13,601 and a mean of 13,991). Subsequent trimming of the first 7 and last 25 bases, as well as reads filtration, led to the generation of sequences with excellent quality (Phred > 30). These high-quality sequences, totaling 258,432 (with a median read count of 9,505 and a mean of 9,940), were subjected to a denoising algorithm[70]. This algorithm effectively merged the overlapping regions R1 and R2, while eliminating chimeric reads.

Taxonomic assignment was carried out using the BLAST feature-classifier, which performed local alignment via BLAST+ between query and reference reads. The consensus taxonomy for each query sequence was assigned based on the latest Greengenes database version (gg_12_10).

For phylogenetic analysis, a rooted tree was constructed utilizing the IQ-TREE stochastic algorithm, facilitating maximum likelihood analysis of extensive phylogenetic data[71].

## Statistics
Experimental group comparisons were performed by applying the nonparametric Mann-Whitney-U (two-tailed MWU) test. To test differences between the developments of the body weight over time in the offspring, linear regression models were built and compared by employing an ANCOVA. For survival comparisons, a log-rank-test (MantelCox-Test) was performed. $P$-values of <0.05 were judged to be significant (*$P < 0.05$, **$P < 0.01$, ***$P < 0.005$, ****$P < 0.001$), exact $P$-values are displayed. Statistical analyzes were conducted using GraphPad Prism version 9.5.1. To discern potential differences in microbiota composition between the groups of subjects (WN *versus* MN and MN *versus* MN+S100a8), the abundance of 16S rRNA gene sequencing reads were normalized by DESeq2, which is based on the negative binomial distribution method (R/Bioconductor DESeq2 package). Then, the LEfSe (Linear discriminant analysis effect size) algorithm[72] was employed for the identification of specific bacterial taxa that could serve as discriminatory markers between the groups.

## Reporting summary
Further information on research design is available in the Nature Portfolio Reporting Summary linked to this article.

## Data availability
16S rRNA sequencing files were submitted to the NCBI Sequence Read Archive (www.ncbi.nlm.nih.gov/sra) and are available with BioProject accession number PRJEB68326 and PRJEB67750. Source data are provided with this paper.

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

## Acknowledgements

We thank Sara Maffei and the staff of the animal facility of the Institute for Research in Biomedicine for excellent mouse husbandry, David Jarossay (Institute for Research in Biomedicine) for cell sorting, and Siegfried Hapfelmeier (University of Bern) for providing the tetracycline-resistant strain of *C. rodentium* used in this study. This work was supported by grants from the Bill & Melinda Gates Foundation in the frame of the project INV-004078 to D.V. and F.G. Further support was provided to D.V. and T.V. from the Federal Ministry of Education and Research (PROSPER; 01EK2103A and 01EK2103D), to D.V. from the Deutsche Forschungsgemeinschaft (DFG, German Research Foundation) in the frame of the projects VI 538/6-3 and VI 538-9-1, SFB 1583/1 ("DECIDE") project number 492620490, TRR 359 ("PILOT") project number 491676693, and under Germany's Excellence Strategy – EXC 2155 'RESIST' – Project ID 390874280, and to F.G. from The Swiss National Science Foundation – grant 310030_192531. J.H. was supported by funds of the Bavarian State Ministry of Science and the Arts and the University of Würzburg to the Graduate School of Life Sciences (GSLS), University of Würzburg.

## Author contributions

L.P., J.H., T.R.J., M.R., S.G., G.G., M.P., T.V., J.R., F.G., and D.V. contributed to the conception, design, and methodology of the study. L.P., J.H., T.R.J., M.R., S.G., G.G., M.P., M.W., B.F., and C.W. performed experiments. J.H. and D.V. wrote the first draft of the manuscript. L.P., P.K.D., T.V., J.R., and F.G. contributed to manuscript revision and editing. All authors read, revised, and approved the manuscript.

## Funding

## Competing interests

F.G. is the founder of MV BioTherapeutics, a company developing mucosal vaccines and biotherapeutics. All remaining authors declare no conflicts of interest.
