## [Peer Review File · Nature Communications]

Postnatal supplementation with alarmins S100a8/a9 ameliorates malnutrition-induced neonate enteropathy in miceREVIEWER EXPERTISE

Reviewer #1. Neonate malnutrition, Microbiome.

Reviewer #2. Alarmins S100A8 and S100A9, Inflammation, Mouse models.

Reviewer #3. Microbiome, Malnutrition, Mouse models.

REVIEWER COMMENTS

Reviewer #1 (Remarks to the Author):

This intriguing manuscript describes the impact of moderate maternal protein and weanling undernutrition on short- and long-term enteropathy in fetal and neonatal mice in the contexts of infection (*C. rodentium* and *S. typhimurium*) and toxin-induced colitis (DSS).

The results shed light on critically important public health challenges. My suggestions for clarifying and improving the manuscript follow:

1. Are any data available on systemic or intestinal IgA induction in the models?
2. More detailed information on the control and MN diets is needed. What was the micronutrient/macronutrient composition? Were the diets isocaloric? Consumption was measured but I did not see where this information is presented. If consumption was different between diets, a pair feeding approach would have strengthened the conclusions.
3. s100a8/a9 binds TLR4. TLR4 activation in the gut induces goblet cell formation. Representative histology from the small intestine and colon would be helpful. Was the abundance of goblet cells influenced by the diet and improved with s100 supplementation? This would provide a potential mechanism.

4. Malnutrition is sometimes associated with coagulopathy, which makes the s100 deficiency interesting in the context of breastmilk and early life.
5. Methods line 359, I believe the correct terminology is "teats" not "tits".
6. EE is both a cause and effect of malnutrition in the view of many.
7. Oral vaccine failure is frequently seen in low-income countries where EE and MN are highly prevalent. It is perhaps beyond the scope of the current manuscript, however I hope the authors will consider testing oral vaccines responses in their models and see whether s100 supplementation might enhance vaccine immunogenicity.

Reviewer #2 (Remarks to the Author):

By using a mouse model, Peruzza et al show that offspring of mothers suffering from protein-energy-malnutrition (PEM) have a dysregulated intestinal immune environment, characterized by a higher influx of neutrophils, higher proportion of inflammatory macrophages and lower infiltration of Tregs. The offspring also had disturbed intestinal flora and developed more aggressive gut inflammation upon bacterial and DSS challenge. The authors link the phenotype to the relative lower S100A8/A9 concentration in breast milk, leading to lower S100A8/A9 concentration in offspring faeces. Feeding the pups one dose of S100A8 reversed the phenotype and is proposed as a potential intervention in offspring of PEM mothers.

Major comments:

- Why has Lcn-2 been chosen as an inflammatory biomarker in this context, over more established biomarkers? Lcn-2 has both pro- and anti-inflammatory properties and it is unclear whether it propagates inflammation or it is produced to counteract excessive inflammation. Which is the cellular source of Lcn-2 in the gut in the context of this work?
- Why is the vast majority of LPMPs CX3CR1(low), even in WN? Are all these MP inflammatory? Why would the intestine of WN control mice contain such high proportions

of inflammatory MP, almost on par with the MN mice? Are the pro- and anti-inflammatory MPs correctly defined in this context?

- Lines 122-123: “Cx3r1...are crucial factors provided by LPMPs...”. CX3CR1 is a receptor and can therefore not be “provided” by MP.

- The differences in S100A8/A9 in breast milk were marginal in the lower concentration range, and only manifested in the higher concentrations range in a few animals. These differences were of approximately the same magnitude in the faeces of offspring on day 7, but the difference was much larger in faeces S100A8/A9 on day 23. Is BM the only S100A8/A9 source in offspring faeces? S100A8/A9 are also produced in high amounts by neutrophils. How can the authors explain that the amount of S100A8/A9 in faeces was lower, despite higher PMN infiltration in the gut?

- The authors observe a similar phenotype when using WN S100A9^{-/-} pups. However, these experiments cannot be used as support for the observed phenotype of pups from PEM mothers. The phenotype of S100A9^{-/-} mice is due to the lack of S100A8/A9 production by the offspring own neutrophils, resulting in an incompetent immune response to bacteria. The mechanisms and the importance of S100A8/A9 in BM in the immediate post-birth period are completely different, as they involve induction of tolerogenic responses to bacterial and inflammatory stimuli. To support their findings, the authors should perform experiments by crossing offspring and feeding S100A9 competent offspring with BM from S100A9^{-/-} mothers, providing the mothers do not reject the “adopted” offspring.

- Why was S100A8 chosen to feed the mice, rather than S100A8/A9 (or S100A9)?

- What happens with S100A8 once it reaches the gut? Does it remain in the gut? Is it absorbed in the circulation? Is it modified by the gut environment? Did the authors check whether S100A8 administration indeed increased S100A8 levels in the gut/faeces and plasma?

- What are the mechanisms/receptors/cell populations that respond to the additional S100A8 administration? The work does not contain any mechanistic studies to explain the described phenotype in the offspring of MN mothers and why S100A8 is beneficial. References 30-32 describe a plethora of possible mechanistic explanations for the importance of gut S100A8/A9 in the early post-birth period. None of these have been explored or discussed by the authors. This represents a major deficiency of the study.

- The “Study limitations” section in the Discussion is missing.

Minor comments:

- The authors use “Cx3r1” in several places in the Results. The correct term is “CX3CR1”.
- The differences between the % of CX3CR1 hi and low LPMPs, although significant, are marginal. Perhaps expressing a low/hi ratio would provide a clearer way of expressing the results.

Reviewer #3 (Remarks to the Author):

This is an interesting and important topic. Physiological and immune adaptational roles of breastmilk S100A8/A9 were previously reported by this group, with altered immunity, intestinal histology and microbiomes in S100-knockout mice which provides a strong underlying rationale for the hypothesis tested here.

The authors clearly demonstrate noteworthy effects of different maternal diets on offspring intestinal structure, function and microbiome mediated by altered breastmilk S100A8/A and priming of the neonatal intestine. The intestinal infection model and amelioration by exogenous S100a8 supplementation provide compelling evidence of effect. However, I am not convinced there is sufficient evidence that this is entirely due to a protein-deficient diet (line 350) and there could be effects occurring independently of maternal malnutrition. Hence more detailed information on the two diets are needed.

Table S2 provides a summary of two commercial chow's crude compositions. Were these regular or purified diets? Besides protein reduction in the ssniff compared to the Altromin product, lipids are significantly reduced, and calorific value is slightly increased by increased carbohydrate. Some lipid species are also critical in regulating inflammation, the microbiome, and lipids may mediate micronutrient bioavailability (e.g. vitamin A which is associated with mucosal immune development) - what was the lipid composition, especially of essential fatty acids, and micronutrient composition of the two feeds? Did they differ in other ways, such as in biologically active non-nutritional compounds? Potential for confounding by these issues is well-documented, for example see:

<https://doi.org/10.1038/labam.1014>

<https://doi.org/10.1093/cdn/nzaa031>

<https://doi.org/10.1007/s11154-019-09512-0>

I question use of the term protein energy malnutrition (PEM), since this fails to recognise the contributions of micronutrients, other nutrients, and inflammation/infection to the syndromes of malnutrition. This terminology was used in the 1970s but now little-used in the human literature since it is recognised that 'PEM' is unlikely to occur in settings where human malnutrition is prevalent. Notably, in this study there was no energy restriction, but other dietary differences.

In the Abstract, please rewrite the sentence 'This resembled the gut phenotype developing upon deficient S100A8/A9 supply via breastmilk and allowed revealing a lack of S100A8/A9 in the breastmilk of malnourished mothers and consequently the offspring is intestine' as it is unclear.

At lines 92/93, '...in the MN offspring compared to the well-nourished (WN) offspring' it was not immediately clear to me whether this referred to the nutritional status of the pups or the dams' nutritional status. Besides, the Methods, section, one sentence here summarising the model, or clarification 'pups fostered by...' would be helpful.

At line 113, 'excluded' is too strong a statement as the comparisons at birth may be limited by sample size. 'Colonic inflammation in the fetus appeared unlikely' is more accurate.

Please discuss the findings reported at lines 198 and 237 in relation to the likely biological half-life of S100A8.

Linked to the first comment, it is important to discuss the need for replication with more refined dietary alterations. Understanding of the biological pathways could also be

strengthen by demonstrating the presence or absence of systemic or intestinal effects of malnutrition in the dams.

Point-by-point response to reviewers

Reviewer #1:

This intriguing manuscript describes the impact of moderate maternal protein and weanling undernutrition on short- and long-term enteropathy in fetal and neonatal mice in the contexts of infection (*C. rodentium* and *S. typhimurium*) and toxin-induced colitis (DSS). The results shed light on critically important public health challenges.

We thank the reviewer for this positive judgment and particularly for the very helpful comments to further improve the manuscript.

1. Are any data available on systemic or intestinal IgA induction in the models?

Answer: New data has been generated to elucidate systemic and intestinal IgA induction in the models. We added the data in the new Figures 2f and 2l and reported them in the Results as follows:

For the *C. rodentium* model: *“In line with the increased bacterial translocation, both intestinal and systemic total IgA levels were lower in MN than WN mice at baseline. Upon infection with C. rodentium systemic IgA was strongly induced in MN mice and became comparable to the levels in infected WN mice. Interestingly, intestinal IgA levels did not significantly change upon C. rodentium infection neither in WN nor MN mice (Fig. 2f).”*

For the *S. typhimurium* model: *“In the S. typhimurium model, the infection increased fecal IgA levels in both groups but in WN mice to higher levels than in MN mice, while systemic IgA levels increased strongly in MN mice but only slightly in WN mice (Fig. 2l).”*

The description of ELISA used for measuring the IgA levels in feces and serum was added to the Methods section.

2. More detailed information on the control and MN diets is needed. What was the micronutrient/macronutrient composition? Were the diets isocaloric? Consumption was measured but I did not see where this information is presented. If consumption was different between diets, a pair feeding approach would have strengthened the conclusions.

Answer: We thank the reviewer for this important notification and agree completely that providing more detailed information on the diets and their consumption is important. Reviewer #3 posed largely the same question (point 1, reviewer #3).

In the revised manuscript, we expanded Supplementary Table 1 (former Supplementary Table 2) to provide more detailed information on the composition of the diets as best as possible. Moreover, we added a new Supplementary Figure 1 illustrating the body weights before start and at termination of diet and daily diet and energy consumption by the dams and revised the Results as follows: *“The calorie contents of the standard and malnutrition diet were similar (Supplementary Table 1) and body weights of MN and WN dams remained comparable until termination of malnutrition (Supplementary Fig. 1a). The proportion of protein from calorie content was 3.9-fold lower in the malnutrition diet, while that of fat was also reduced (2.3-fold) and that of carbohydrates 1.5-fold higher compared to the standard diet (Supplementary Table 1). Interestingly, MN dams consumed more chow and calories per day than well-nourished (WN) dams (Supplementary Fig. 1b,c). Yet the mean energy uptake of MN dams by protein*

was still 3.0-fold reduced compared to WN dams, while that by fat remained 1.8-fold lower and that by carbohydrates 1.9-fold higher (Supplementary Fig. 1d-f).". In the Method section describing the model, we notify the reader now that the malnutrition diet is primarily protein-reduced but also fat-reduced and carbohydrate-enriched.

Acknowledging the multiple differences between the diets, we completely agree with the reviewer that it is better to abandon the term protein-energy-malnutrition (PEM). In the revised manuscript, we restrict to the term "malnutrition" following this explanatory section added to the Results as well in the Abstract.

To notify the readership about the weakness of the model regarding the multiple differences between the standard and protein-reduced diet we added the following section to the Discussion: *"Protein deficiency was the main dietary alteration caused in MN mothers. Albeit to a lesser extent, the diet applied for malnutrition in our model was also deficient in fat and enriched in carbohydrates and additionally purified in contrast to the grain-based standard diet, which collectively might have contributed to the immunological and microbial alterations observed in the offspring of MN mothers⁶⁷. In this context, particularly reduced supply of fatty acids and vitamin A might have contributed to the alteration in intestinal structures and gut flora of the offspring^{68,69}. However, studies that have focused on maternal fatty acid or vitamin A deficiencies differ from our results with respect to the microbiota alterations in the offspring. For example, contrary to what we observed in our model in the offspring of MN dams, maternal deficiency of omega-3 long-chain polyunsaturated fatty acids has been associated with a higher relative abundance of Clostridiaceae⁶⁹ and vitamin A deficiency neither altered the abundance of Enterobacteriaceae nor Clostridiaceae⁶⁸. To clarify to what extent biologically active proteins other than S100A8/A9 or lipid species or vitamins influence the gut phenotype developing under malnutrition follow-up studies with more refined dietary alterations of the mothers are required."*

Given the higher chow consumption in the group of MN dams, a pair feeding approach likely would aggravated the malnutrition effect and might have strengthened the conclusions. However, this would have required multiple daily non-ad-lib gavage feeding which would have represented enormous stress for the mothers and at the same time the offspring. Maternal and infant stress underlies high inter-individual variance and is a well-known factor impacting on the offspring's development in many aspects. We waived this approach as we feared a strong confounder effect coming with the potentially induced uncontrolled stress.

3. s100a8/a9 binds TLR4. TLR4 activation in the gut induces goblet cell formation. Representative histology from the small intestine and colon would be helpful. Was the abundance of goblet cells influenced by the diet and improved with s100 supplementation? This would provide a potential mechanism.

Answer: We thank the reviewer for raising further considerations on the mechanisms of how S100a8/a9 ensures gut homeostasis. The reviewer is right that S100a8/a9 binds and activates TLR4 signaling (Vogl et al., 2007; Fassl et al., 2014 (PMID: 25505274), Austermann et al. 2014; Möller et al., 2023 (PMID: 37056775)). The effect of TLR4 signaling in the gut is complicated and often appears contradictory, which is linked to the fact that the effect is a question of signaling balance as well as setting and timing of assessment in models. Continuous TLR4 signaling under steady-state conditions - either activated by S100a8/a9 (Willers et al., 2020) or LPS derived from the microbiota (Rakoff-Nahoum et al., 2004 (PMID: 15260992)) - is essential for the establishment of gut homeostasis. Contrary, exaggerated

TLR4 signaling has been linked to an increased susceptibility to inflammatory bowel disease (IBD) in adults (Hausmann et al., 2020 (PMID: 12055604); Singh et al., 2005 (PMID: 15499080)) and necrotizing enterocolitis (NEC) in neonates (Sodhi et al., 2012 (PMID: 22796522); McElroy et al., 2011 (PMID: 21737776)). Both seemingly opposite effects of TLR4 signaling have been further supported by chimeric mice and TLR4 inhibition studies showing that blocking TLR4 ameliorates inflammation but impairs mucosal healing in murine colitis (Ungaro et al., 2009 (PMID: 19359427)). We believe that in inflammatory disease settings it is not yet fully understood whether TLR4 signaling has only a pathogenetic role or is essentially induced in response to the injury in order to initiate the process of repair and reestablishment of homeostasis in the intestinal tissue niche. Seen this way, its primary function would be the same in adults and neonates, i.e. reestablishing gut homeostasis after injury respective setting up gut homeostasis after birth.

Specifically, with respect to its effect on Goblet cells, most studies report that TLR4 signaling rather regulates than induces Goblet cell formation (Sodhi et al., 2012 (PMID: 22796522), Lanik et al., 2023 (PMID: 36881475)). In the intestine of neonates with NEC, TLR4 signaling is increased and Goblet cell numbers are reduced (Sodhi et al., 2012; McElroy et al., 2011 (PMID: 21737776); Chaaban et al., 2022 (PMID: 35336095)). S100a8/a9 activates TLR4 signaling but we never observed enhanced Goblet cell formation in the *S100a9*^{-/-} mouse (e.g., Willers et al., 2020). Nevertheless, we checked Goblet cell abundance in the SI and LI of WN and MN mice but found no differences (Figure R1). We decided not to add this new data to the revised manuscript since this would require an introductory explanation why we performed these studies. Such explanation would basically say “To test whether S100a8/a9 deficiency in MN mice and consequently reduced S100a8/a9-mediated TLR4 signaling leads to enhanced Goblet cell formation....”, which is likely confusing for the readership as Goblet cell formation is protective as outlined above (Sodhi et al., 2012; McElroy et al., 2011 (PMID: 21737776); Chaaban et al., 2022 (PMID: 35336095)).

Fig. R1 Comparable abundance of goblet cells in the SI of WN and MN mice. Representative histology images of the SI of d23 WN (left) and MN (right) mice. There is a comparable abundance of goblet cells (arrowheads) visible in PAS-reaction. Additionally, the enterocytes display apical cytoplasmic vacuoles, that are negative in the PAS-reaction (arrow). Scale bars, 50 μ m.

[Method: After harvesting, intestines were fixed in 4% buffered formalin. Specimens were embedded in paraffin and 2 μ m thick sections were cut followed by histological staining using periodic acidic-Schiff reaction to highlight goblet cells. The histological analysis was performed blinded to experimental setup on a routine diagnostic light microscope (BX43, Olympus, Tokyo, Japan). Representative images were acquired with an Olympus CS50 camera (Olympus, Tokyo, Japan) using Olympus cellSens Software (Olympus, Tokyo, Japan).]

Given our finding of comparable Goblet cell numbers, one could still ask why S100a8/a9 deficiency in MN mice does not cause enhanced Goblet cell formation. A possible explanation might be the overexpansion of *Enterobacteriaceae* in the colonizing microbiota in S100a8/a9 deficient settings like in the model of this study or in *S100a9*^{-/-} murine neonates or preterm

infants (Willers et al., 2020). The high abundance of *Enterobacteriaceae* might provide bacterial cell wall LPS that maintains TLR4 signaling.

However, there is another interesting aspect that could be relevant in the context of S100A8/A9 influencing Goblet cell functions. By using intravital two-photon microscopy, the group of Rodney Newberry observed that EGF from breast milk leads to reduced formation of Goblet cell-associated antigen passages (GAPs) and decreased *E. coli* translocation from the murine intestine (Knoop et al., 2020 (PMID: 32179676)). More previously they showed that the regulation of GAP formation is dependent on TLR4-MyD88-mediated microbial sensing, particularly LPS-TLR4-MyD88 signaling (Knoop et al., 2015 (PMID: 25005358)). This has been corroborated by another group showing that TLR4 signaling is essential for limiting bacterial translocation in a murine model of colitis (Fukata et al., 2005 (PMID: 15826931)). We added the aspect of a possible additional impact of S100A8/A9 on the intestinal epithelium to the discussion: "*S100A8/A9 binds and activates TLR4 signaling*^{43,48,49,60}. *In neonates, continuous S100A8/A9-TLR4-signaling tolerizes blood monocytes*³¹ *and induces a regulatory phenotype in intestinal LPMPs, which promotes the expansion of Tregs and is associated with a controlled expansion of Gammaproteobacteria like Escherichia coli*²⁹. *The latter overgrows likewise in the gut of TLR4 knockout mice*⁶¹. *Continuous TLR4 signaling is also known to regulate the proliferation and differentiation of the intestinal epithelium*^{62,63} *and limit bacterial translocation, e.g. through restricting the formation of Goblet cell-associated antigen passages*⁶⁴⁻⁶⁶. *Continuous S100A8/A9-TLR4 signaling in the neonatal gut might therefore also protect from dysbiosis and bacterial translocation by impacting on the intestinal epithelium which future studies must validate experimentally*

4. Malnutrition is sometimes associated with coagulopathy, which makes the s100 deficiency interesting in the context of breastmilk and early life.

Answer: This is an interesting note. In our work program, we have not included platelet or coagulation studies. At least clinically, neither the dams nor the offspring showed signs of purpura and the histology preparations from intestinal tissue samples showed no signs of disseminated intravascular coagulation (DIC).

5. Methods line 359, I believe the correct terminology is "teats" not "tits".

Answer: The reviewer is right; this was a mistake and we corrected it.

6. EE is both a cause and effect of malnutrition in the view of many.

Answer: We completely agree with the reviewer's note! We were not certain whether the reviewer wanted us to stress this better. We added this important notification to the Introduction: "*Thus, EE is both an effect and a cause of malnutrition.*".

7. Oral vaccine failure is frequently seen in low-income countries where EE and MN are highly prevalent. It is perhaps beyond the scope of the current manuscript, however I hope the authors will consider testing oral vaccines responses in their models and see whether s100 supplementation might enhance vaccine immunogenicity.

Answer: This is an excellent suggestion of the reviewer and encourages us to proceed! We started already to look in our human term and preterm infant birth cohorts (though

malnourished) whether the response to rotavirus vaccine is dependent on S100 supply via breast milk.

Reviewer #2:

By using a mouse model, Peruzza et al show that offspring of mothers suffering from protein-energy-malnutrition (PEM) have a dysregulated intestinal immune environment, characterized by a higher influx of neutrophils, higher proportion of inflammatory macrophages and lower infiltration of Tregs. The offspring also had disturbed intestinal flora and developed more aggressive gut inflammation upon bacterial and DSS challenge. The authors link the phenotype to the relative lower S100A8/A9 concentration in breast milk, leading to lower S100A8/A9 concentration in offspring faeces. Feeding the pups one dose of S100A8 reversed the phenotype and is proposed as a potential intervention in offspring of PEM mothers.

Major comments:

1. Why has Lcn-2 been chosen as an inflammatory biomarker in this context, over more established biomarkers? Lcn-2 has both pro- and anti-inflammatory properties and it is unclear whether it propagates inflammation or it is produced to counteract excessive inflammation. Which is the cellular source of Lcn-2 in the gut in the context of this work?

Answer: Different arguments addressing to different aspects of this work and our models led to the decision of choosing Lcn-2 as marker of inflammation:

- i. Fecal Lcn-2 requires only a stool sample, thus permitting measurement over time.
- ii. The group of Emma Slack uses the same *Salmonella* infection model as we used in this work and always measured Lcn-2 in the feces to quantify intestinal inflammation (Diard et al., 2021 (PMID: 34045711); Pfister et al., 2020 (PMID: 32332737); Moor et al., 2017 (PMID: 28405025)) wherefore we considered Lcn-2 as a well-established marker of inflammation.
- iii. Previous studies demonstrated that Lcn-2 has the sensitivity to detect low-grade intestinal inflammation while having a dynamic range broad enough to reflect classic robust intestinal inflammation (Chassaing et al., 2012 (PMID: 22957064)). This made Lcn-2 particularly attractive in our work which was aiming at detecting persistent low-grade intestinal inflammatory gut dysfunction as well as enteric infection-mediated strong inflammatory intestinal responses.
- iv. Fecal as well as systemic S100A8/A9 (calprotectin), both sensitive biomarkers of inflammation, do not serve as intestinal respective systemic inflammatory biomarker in neonates due to high baseline levels in healthy neonates, in humans as well as in mice (Willers et al., 2020; Ulas et al., 2017; Austermann et al., 2014). Therefore, we have been glad when we could confirm in our preparatory studies that Lcn-2 serves as a sensitive marker dissecting between neonates with healthy gut states and neonates with inflammatory intestinal conditions. Meanwhile, also other groups discovered Lcn-2 as marker that is apt to detect low-grade inflammation in neonates (Gravina et al., 2023 (PMID: 37496672)).

We completely agree with the reviewer that Lcn-2, has both pro- and anti-inflammatory properties (like S100A8/A9 in dependence of age (Viemann 2020)). Thus, it might propagate (classical alarmin function) but also counteract inflammation. However, it was beyond the scope of this work to clarify which was the cellular source of Lcn-2 and the role of Lcn-2 in the gut in the context of this model. Here, we used Lcn-2 just as biomarker.

2. Why is the vast majority of LPMPs CX3CR1(low), even in WN? Are all these MP inflammatory? Why would the intestine of WN control mice contain such high proportions of inflammatory MP, almost on par with the MN mice? Are the pro- and anti-inflammatory MPs correctly defined in this context?

Answer: Only a few studies exist which report the proportions of Cx3cr1^{low} and Cx3cr1^{hi} LPMP subsets in the intestine, especially not over age and not at all in the small intestine. In this study, report for the first time on developmental changes in murine LPMP subset distributions in the SI. With respect to the LI, we actually could recapitulate numbers and proportions of LPMP subsets in WN mice published previously. Bain et al., 2014 and our group (Willers et al., 2020) showed that LPMPs in the LI of murine neonates are F4/80^{hi} yolk-sac derived LPMPs with virtually all of them being Cx3cr1^{hi} LPMPs (98% Cx3cr1^{hi} LPMPs / 2% Cx3cr1^{low} LPMPs). After birth, they are replaced by hematopoietic blood-derived F4/80^{low} LPMPs which are Cx3cr1 low expressing and locally further differentiate into Cx3cr1 high expressing tissue resident LPMPs if the niche of tissue resident LPMPs needs further replenishment (Bain et al., 2014; Bain et al., 2013). Thus, after birth a new equilibrium is established over time reaching an adult ratio of approximately 3% Cx3cr1^{hi} LPMPs / 70% Cx3cr1^{low} LPMPs at d21 of life (Willers et al., 2020; Bain et al., 2014). Consequently, the expression of Cx3cr1 on intestinal LPMPs is a continuum and the lower Cx3cr1 is expressed on Cx3cr1^{low} LPMPs the more recently they have been recruited from the blood and the better they might mark the proinflammatory subsets among Cx3cr1^{low} LPMPs.

Therefore, we analyzed the mean expression of Cx3cr1 on the Cx3cr1^{low} LPMPs of WN and MN mice (new Figure 1i). In both SI and LI, the mean expression of Cx3cr1 on Cx3cr1^{low} LPMPs of d23 MN mice was significantly lower than in WN mice, suggesting that they represent only recently immigrated blood-derived Cx3cr1^{low} LPMPs and corroborating their proinflammatory nature (Bain et al., 2013) as validated by the transcriptomic studies (Fig. 1k (former Fig. 1j)).

3. Lines 122-123: "Cx3cr1...are crucial factors provided by LPMPs...". CX3CR1 is a receptor and can therefore not be "provided" by MP.

Answer: We thank the reviewer for this hint and changed the wording to "*Cx3cr1 along with Il-10 and Tgf-β are crucial factors that trigger the postnatal expansion of regulatory T cells (Tregs) in the gut mucosa. In line with their reduced expression by LPMPs, ...*".

4. The differences in S100A8/A9 in breast milk were marginal in the lower concentration range, and only manifested in the higher concentrations range in a few animals. These differences were of approximately the same magnitude in the faeces of offspring on day 7, but the difference was much larger in faeces S100A8/A9 on day 23. Is BM the only S100A8/A9 source in offspring faeces? S100A8/A9 are also produced in high amounts by neutrophils. How can the authors explain that the amount of S100A8/A9 in faeces was lower, despite higher PMN infiltration in the gut?

Answer: As we showed earlier, breast milk shows the highest levels of S100A8/A9 hitherto measured in biomaterials and is the most relevant source of S100A8/A9 levels in the neonatal gut (Pirr et al., 2017). In an ongoing clinical study in human birth cohorts, we see a strict correlation between S100A8/A9 concentrations in breast milk and fecal samples (unpublished data). This is in line with the human data of Savino et al. 2010 (PMID: 19887860) and Lee et

al. 2017 (PMID: 28391117) who found higher fecal S100A8/A9 levels in breast milk-fed infants compared to formula-fed infants. However, the reviewer is right that offspring-derived S100A8/A9 produced by myeloid cells including LPMPs and PMNs can contribute to the S100A8/A9 levels in the neonatal gut as shown previously (Willers et al., 2020).

We agree with the reviewer that S100a8/a9 in murine breast milk varied considerably but still they clearly differed significantly between WN and MN dams. Collection of breast milk during the first week after delivery in murine mothers is more challenging than in human mothers, which certainly contributes to the variance. Therefore, we collected large numbers of breast milk samples (WN: $n=49$, MN: $n=63$) to generate robust data.

We do not find the differences in fecal S100a8/a9 levels of the offspring being approximately the same magnitude on day 7. They are significantly different with a mean fS100a8/a9 level in d7 WN mice of 1,253 ng/ml (SD \pm 911) in contrast to 432 ng/ml (SD \pm 226) in d7 MN mice. Yet, the reviewer is right that the difference between both groups in fecal S100a8/a9 levels was more significant at d23. Therefore, the reviewer's suggestion that the higher PMN infiltration in the gut at day 7 might represent a significant source for fecal S100a8/a9 in the MN group could indeed be a good explanation that the difference between fecal S100a8/a9 in d7 WN and MN mice is diminished compared to d23. To substantiate this hypothesis, we performed new S100a9 immunostainings of d7 PMNs in LI tissue samples to exclude S100 expression deficits in PMNs from MN mice and demonstrate PMN infiltrates in the LI of MN mice, whereas PMNs in the LI of WN mice were rare and difficult to detect. We added this data as new Fig. 1f and related to them additionally in the third Result section as follows: *"Subsequent quantification in breast milk, the most important source of fecal S100a8/a9 for the infant^{29,42}, revealed significantly lower S100a8/a9 levels in MN compared to WN dams (Fig. 3d). Fecal S100a8/a9 levels in the offspring did not yet differ at birth (d1) but became lower in MN mice from d7 on and remained strongly decreased compared to WN mice even at d23 after weaning. S100a8/a9 deficiency of MN mice at d7 might have been mitigated by the infiltrating PMNs, which are well-known producers of S100a8/a9 (Fig. 1e,f)."*

The Methods section was extended accordingly by the description of Immunofluorescence microscopy.

5. The authors observe a similar phenotype when using WN S100A9^{-/-} pups. However, these experiments cannot be used as support for the observed phenotype of pups from PEM mothers. The phenotype of S100A9^{-/-} mice is due to the lack of S100A8/A9 production by the offspring own neutrophils, resulting in an incompetent immune response to bacteria. The mechanisms and the importance of S100A8/A9 in BM in the immediate post-birth period are completely different, as they involve induction of tolerogenic responses to bacterial and inflammatory stimuli. To support their findings, the authors should perform experiments by crossing offspring and feeding S100A9 competent offspring with BM from S100A9^{-/-} mothers, providing the mothers do not reject the "adopted" offspring.

Answer: The reviewer is right in pointing out that S100a8/a9 might contribute to host defense by antimicrobial effects. We thank the reviewer for making us aware that this important gap in our line of argumentation had to be clarified.

Only the heterodimer S100a8/a9 exerts bacteriostatic effects by chelating of Mn²⁺ and Zn²⁺ (Corbin et al., 2008 (PMID: 18276893); Achouiti et al., 2012 (PMID: 23133376); Pirr et al., 2017). Moreover, the bacteriostatic effect is only achieved at high concentrations (starting at 30 μ g/ml) as found in tissue abscesses (Corbin et al., 2008) since each S100-molecule

contains only two binding sites for divalent metal ions. In addition, the antimicrobial effect of S100a8/a9 is strongly dependent on the setting and bacterial strain. Under some conditions, depletion of metal ions by S100a8/a9 can also enhance the virulence and growth of bacteria (Achouiti et al., 2014 (PMID: 25179663); Cho et al., 2015 (PMID: 26147796)). To clarify whether S100a8/a9 exerts antimicrobial activity towards the *C. rodentium* strain and the *S. typhimurium* strain used in our models we performed new experiments and found that the presence of S100a8/a9 inhibited neither the growth of *C. rodentium* nor *S. typhimurium* significantly. We added the new data to Supplementary Figure 4i (former Supplementary Figure 3) and report them in the third paragraph of Results as follows: “*Direct antimicrobial effects of S100a8/a9 against C. rodentium and S. typhimurium could be excluded (Supplementary Figure 4i), supporting that S100a8/a9 plays an important immunoregulatory role that protects against the development of enteropathy and susceptibility to enteric infections under maternal malnutrition.*”. The description of the bacterial growth inhibition assay has been added to the Methods.

Besides the striking overlap with the phenotype of MN pups, we find the novel findings on the phenotype of *S100a9*^{-/-} mice in the enteric infection models very intriguing and worth to report. They support the important immunoregulatory role of S100a8/a9 in the context of enteric infections and thus contribute to better understanding of host pathogen interactions.

Crossing offspring and feeding S100a9 competent offspring with BM from *S100a9*^{-/-} mothers has already been performed previously by our group (Willers et al., 2020). These experiments demonstrated that breast milk-derived S100a8/a9 is essential while pup-derived S100a8/a9 produced by myeloid cells was not sufficient to achieve the full establishment of gut homeostasis.

6. Why was S100A8 chosen to feed the mice, rather than S100A8/A9 (or S100A9)?

Answer: Two important aspects let us chose S100a8 (alias myeloid related protein 8 (MRP8)). First, S100a8 proved in several previous studies as the most bioactive S100 formulation with respect to immunoregulatory effects in comparison to S100a9 (alias MRP14) and S100a8/a9 (alias MRP8/MRP14 or calprotectin) (Austermann et al., 2014; Heinemann et al., 2017; Ulas 2017). Moreover, with respect to the impact of its immunoregulatory effects on the development of the gut microbiome we wanted to avoid objections due to potential antimicrobial effects of S100a8/a9 (see our answer to point 5) and went for a supply of S100a8. The homodimers S100a8 or S100a9 exert no antimicrobial effects since they do not have transition binding sites for divalent metal ions (Pirr et al., 2017; Damo et al., 2013 (PMID: 23431180); Achouiti et al., 2012 (PMID: 23133376); Corbin et al., 2008 (PMID: 18276893); Vogl et al., 2006 (PMID: 17050004)). In order to provide the reader with the aforementioned data situation we added this important information in the fourth Result section as follows: “*S100a8 homodimers were chosen as they are the most immunoactive form of S100-alarmins^{31,43,44} and have no antimicrobial effects (Supplementary Figure 4i and ^{42,45,46}).*”.

7. What happens with S100A8 once it reaches the gut? Does it remain in the gut? Is it absorbed in the circulation? Is it modified by the gut environment? Did the authors check whether S100A8 administration indeed increased S100A8 levels in the gut/faeces and plasma?

Answer: We thank the reviewer for this important question. Unfortunately, there is no quantitative assay available for measuring murine S100a8 homodimers. Our in-house as well

as commercial ELISA assays only quantify the heterodimer S100A8/A9. However, semi-quantitative analysis by western blot is possible and has been done previously (Pirr et al., 2019) showing retrieval in the plasma of S100ko mice fed with S100a8/a9 and S100a9, respectively. In the gut, we had not yet tried to retrieve supplemented S100a8 and therefore performed new experiments for immunoblotting S100a8 using fecal sample extracts from MN mice 8h, 3 days and 7 days after feeding PBS or S100a8 (new Supplementary Fig. 5b,c). In the revised manuscript, this new data is reported in the fourth Result section: “*Supplemented S100a8 could be retrieved in the feces of MN mice for up to 3 days after feeding (Supplementary Fig. 5b,c), consistent with the extracellular half-life of S100a8/a9 of approximately 24 hours^{47,48}.*”. The description of Immunoblotting has been added to the Methods section.

8. What are the mechanisms/receptors/cell populations that respond to the additional S100A8 administration? The work does not contain any mechanistic studies to explain the described phenotype in the offspring of MN mothers and why S100A8 is beneficial. References 30-32 describe a plethora of possible mechanistic explanations for the importance of gut S100A8/A9 in the early post-birth period. None of these have been explored or discussed by the authors. This represents a major deficiency of the study.

Answer: Regarding mechanisms and receptors that mediate the S100A8 effects we kindly ask to refer to previously published work of our group where we demonstrated TLR4 dependence of S100 effects in different models (Vogl et al., 2007; Fassl et al., 2014 (PMID: 25505274); Austermann et al. 2014; Möller et al., 2023 (PMID: 37056775)). Reviewer #1 considers TLR4 being the receptor of S100a8/a9 to be set. It would be beyond the scope of this work to verify specifically in this MN model again that TLR4 is the receptor mediating the immunoregulatory effects of S100a8/a9. This would require additional large experimental series of WN, MN and S100-MN settings in TLR4-knockout mice. However, the reviewer’s question prompted us to verify the Tlr4 expression status in the intestine of MN mice in comparison to WN mice using Immunofluorescence microscopy (new Supplementary Fig. 5a). As anticipated from the treatment success of S100a8 supplementation in MN mice (Fig. 4-6), Tlr4 expression in the MN intestine was not impaired. We added the following sentence to the result section: “*The intestinal expression of Tlr4, the major receptor of S100a8/a9^{43,47-49}, was comparable in the WN and MN offspring (Supplementary Fig. 5a).*”. The Methods section was extended accordingly.

With respect to mechanistic insights, we provided data on the reprogramming of the LPMP phenotype and related Treg expansion upon S100a8 supplementation. However, the reviewer is right that further mechanistic insights especially regarding the impact on S100a8 on the intestinal epithelium would be very interesting as IEC damage is for instance a likely major trigger for the early PMN infiltration and prevented by S100a8 supplementation. We intend to deepen our studies on how exactly and to what extent fS100A8/A9 impacts on the intestinal epithelium (actually a new project addressing to this question started already at the beginning of this year). Herein, thorough mechanistic studies are planned including advanced tissue models (e.g. organoid cultures from mouse SI and LI), animal models with targeted TLR4 knockouts and *in vivo* life-imaging.

Reviewer #1 was also wondering on the role of TLR4 signaling on the formation and function of goblet cells (point 3). Therefore, we added the aspect of a possible additional impact of S100A8/A9 on the intestinal epithelium to the revised discussion, thereby also including the

mechanistic aspects based on (former) References 30-32 as follows: “S100A8/A9 binds and activates TLR4 signaling^{43,48,49,60}. In neonates, continuous S100A8/A9-TLR4-signaling tolerizes blood monocytes³¹ and induces a regulatory phenotype in intestinal LPMPs, which promotes the expansion of Tregs and is associated with a controlled expansion of Gammaproteobacteria like *Escherichia coli*²⁹. The latter overgrows likewise in the gut of TLR4 knockout mice⁶¹. Continuous TLR4 signaling is also known to regulate the proliferation and differentiation of the intestinal epithelium^{62,63} and limit bacterial translocation, e.g. through restricting the formation of Goblet cell-associated antigen passages^{64–66}. Continuous S100A8/A9-TLR4 signaling in the neonatal gut might therefore also protect from dysbiosis and bacterial translocation by impacting on the intestinal epithelium which future studies must validate experimentally.”.

9. The “Study limitations” section in the Discussion is missing.

Answer: We thank the reviewer for this hint. The major study limitations are now extensively pointed out at suitable points in the revised Discussion. These are in particular the need of further studies giving deeper mechanistic insights on the mode of action of S100a8/a9 (as outlined in our answer to point 8) and that not PEM-related dietary influences cannot be fully excluded (please see Reviewer #1/point 2 and Reviewer #3/point 1).

Minor comments:

10. The authors use “Cx3cr1” in several places in the Results. The correct term is “CX3CR1”.

Answer: We apologize this careless mistake and corrected it throughout the manuscript.

11. The differences between the % of CX3CR1 hi and low LPMPs, although significant, are marginal. Perhaps expressing a low/hi ratio would provide a clearer way of expressing the results.

Answer: Good suggestion! Unfortunately, the plotting of ratios did not substantially improve the illustration of the significant differences between WN, MN and S100-supplemented MN mice as shown in Figure R2 (for the reviewer). We therefore would like to keep reporting the proportions of Cx3cr1^{hi} and Cx3cr1^{low} LPMPs in Figure 1 and Figure 4.

Fig. R2 Balance between proinflammatory Cx3cr1^{low} LPMP and regulatory Cx3cr1^{hi} LPMP in WN and MN mice. a Ratio of Cx3cr1^{low} to Cx3cr1^{high} LPMPs in WN and MN mice at d7 (n=7-12 each group) and d23 (n=12-15 each group). **b** Ratio of Cx3cr1^{low} to Cx3cr1^{high} LPMPs in d23 MN mice fed PBS (MN) and S100a8 (MN+S100a8) after birth (n=10-11 each group). Bars represent means ± SEM. *p < 0.05, **p < 0.01, ****p < 0.0001, n.s., not significant (MWU-tests).

Reviewer #3:

This is an interesting and important topic. Physiological and immune adaptational roles of breastmilk S100A8/A9 were previously reported by this group, with altered immunity, intestinal histology and microbiomes in S100-knockout mice which provides a strong underlying rationale for the hypothesis tested here.

1. The authors clearly demonstrate noteworthy effects of different maternal diets on offspring intestinal structure, function and microbiome mediated by altered breastmilk S100A8/A and priming of the neonatal intestine. The intestinal infection model and amelioration by exogenous S100a8 supplementation provide compelling evidence of effect. However, I am not convinced there is sufficient evidence that this is entirely due to a protein-deficient diet (line 350) and there could be effects occurring independently of maternal malnutrition. Hence more detailed information on the two diets are needed.

Table S2 provides a summary of two commercial chow's crude compositions. Were these regular or purified diets? Besides protein reduction in the ssniff compared to the Altromin product, lipids are significantly reduced, and calorific value is slightly increased by increased carbohydrate. Some lipid species are also critical in regulating inflammation, the microbiome, and lipids may mediate micronutrient bioavailability (e.g. vitamin A which is associated with mucosal immune development) - what was the lipid composition, especially of essential fatty acids, and micronutrient composition of the two feeds? Did they differ in other ways, such as in biologically active non-nutritional compounds? Potential for confounding by these issues is well-documented, for example see:

<https://doi.org/10.1038/labam.1014>

<https://doi.org/10.1093/cdn/nzaa031>

<https://doi.org/10.1007/s11154-019-09512-0>

I question use of the term protein energy malnutrition (PEM), since this fails to recognise the contributions of micronutrients, other nutrients, and inflammation/infection to the syndromes of malnutrition. This terminology was used in the 1970s but now little-used in the human literature since it is recognised that 'PEM' is unlikely to occur in settings where human malnutrition is prevalent. Notably, in this study there was no energy restriction, but other dietary differences.

Answer: We thank the reviewer for this question and particularly for the further explanations how critical the choice and evaluation of rodent diets is. This helped us a lot to better elaborate the differences between the diets and to improve the manuscript. Reviewer #1 posed largely the same question (point 2, reviewer #1).

In the revised manuscript, we expanded Supplementary Table 1 (former Supplementary Table 2) to provide more detailed information on the composition of the diets as best as possible. Moreover, we added a new Supplementary Figure 1 illustrating the body weights before start and at termination of diet and daily diet and energy consumption by the dams and revised the Results as follows: *"The calorie contents of the standard and malnutrition diet were similar (Supplementary Table 1) and body weights of MN and WN dams remained comparable until termination of malnutrition (Supplementary Fig. 1a). The proportion of protein from calorie content was 3.9-fold lower in the malnutrition diet, while that of fat was also reduced (2.3-fold) and that of carbohydrates 1.5-fold higher compared to the standard diet (Supplementary Table 1). Interestingly, MN dams consumed more chow and calories per day than well-nourished (WN) dams (Supplementary Fig. 1b,c). Yet the mean energy uptake of MN dams by protein was still 3.0-fold reduced compared to WN dams, while that by fat remained 1.8-fold lower and that by carbohydrates 1.9-fold higher (Supplementary Fig. 1d-f)."* In the Method section

describing the model, we also notify the reader now that the malnutrition diet used in our model is primarily protein-reduced but also fat-reduced and carbohydrate-enriched.

Acknowledging the multiple differences between the diets, we completely agree with the reviewer that it is better to abandon the term protein-energy-malnutrition (PEM). In the revised manuscript, we restrict to the term “malnutrition” following this explanatory section added to the Results as well in the Abstract.

To notify the readership about the weakness of the model regarding the multiple differences between the standard and malnutrition diet we added the following section to the Discussion: *“Protein deficiency was the main dietary alteration caused in MN mothers. Albeit to a lesser extent, the diet applied for malnutrition in our model was also deficient in fat and enriched in carbohydrates and additionally purified in contrast to the grain-based standard diet, which collectively might have contributed to the immunological and microbial alterations observed in the offspring of MN mothers⁶⁷. In this context, particularly reduced supply of fatty acids and vitamin A might have contributed to the alteration in intestinal structures and gut flora of the offspring^{68,69}. However, studies that have focused on maternal fatty acid or vitamin A deficiencies differ from our results with respect to the microbiota alterations in the offspring. For example, contrary to what we observed in our model in the offspring of MN dams, maternal deficiency of omega-3 long-chain polyunsaturated fatty acids has been associated with a higher relative abundance of Clostridiaceae⁶⁹ and vitamin A deficiency neither altered the abundance of Enterobacteriaceae nor Clostridiaceae⁶⁸. To clarify to what extent biologically active proteins other than S100A8/A9 or lipid species or vitamins influence the gut phenotype developing under malnutrition follow-up studies with more refined dietary alterations of the mothers are required.*

2. In the Abstract, please rewrite the sentence ‘This resembled the gut phenotype developing upon deficient S100A8/A9 supply via breastmilk and allowed revealing a lack of S100A8/A9 in the breastmilk of malnourished mothers and consequently the offspring is intestine’ as it is unclear.

Answer: We are glad that the reviewer requests clarification as we were never really happy with this sentence. We now tried to improve it in the frame of word restriction and changed it as follows: *“This gut phenotype resembled those developing when S100a8/a9-supply via breast milk is deficient. We could confirm that S100a8/a9 is lacking in the breast milk of malnourished mothers and the offspring’s intestine.”.*

3. At lines 92/93, ‘...in the MN offspring compared to the well-nourished (WN) offspring’ it was not immediately clear to me whether this referred to the nutritional status of the pups or the dams’ nutritional status. Besides, the Methods, section, one sentence here summarising the model, or clarification ‘pups fostered by...’ would be helpful.

Answer: We agree with the reviewer that the terminology can be confusing and that a definition of terms would be helpful for the readership and precision. In the revised manuscript, we added such definition: *“In the offspring, the induction of stunting and enteropathy was verified in d23 mice of MN compared to WN dams (in the following named MN respective WN mice or pups) by....”.* In the Method section we added the following sentence: *“The offspring was named WN respective MN pups or mice when fostered by WN respective MN dams.”.*

4. At line 113, 'excluded' is too strong a statement as the comparisons at birth may be limited by sample size. 'Colonic inflammation in the fetus appeared unlikely' is more accurate.

Answer: The reviewer is right. We are grateful for this constructive suggestion. We could increase a bit the sample size of d1 MN mice (4 more) and also changed the sentence as suggested by the reviewer.

5. Please discuss the findings reported at lines 198 and 237 in relation to the likely biological half-life of S100A8.

Answer: The *in vivo* extracellular half-life of S100a8/a9 in humans and mice is approximately 24 hours (Viemann et al., 2005 (PMID: 15598812); Hirono et al., 2006 (PMID: 16979015); van Zoelen et al., 2009 (PMID: 19762566)). The half-life of S100a8 has not been determined yet. As Reviewer #2 also asked what happens with the S100a8 after supplementation, we performed retrieval experiments and report the new data in the fourth Result section: "*Supplemented S100a8 could be retrieved in the feces of MN mice for up to 3 days after feeding (Supplementary Fig. 5b,c), consistent with the extracellular half-life of S100a8/a9 of approximately 24 hours^{47,48}.*". Please, see also our answer to Reviewer #2/point 7.

6. Linked to the first comment, it is important to discuss the need for replication with more refined dietary alterations. Understanding of the biological pathways could also be strengthened by demonstrating the presence or absence of systemic or intestinal effects of malnutrition in the dams.

Answer: This is also a very helpful comment. We added this important notification for the readership to the Discussion as outlined in our answer to the first comment.

REVIEWERS' COMMENTS

Reviewer #1 (Remarks to the Author):

The authors have thoroughly addressed my concerns. No further suggestions.

Reviewer #2 (Remarks to the Author):

I would like to thank the authors for the thorough answer to my previous comments. The additional experiments performed have increased the value and the clarity of the findings. I have no further comments.

Reviewer #3 (Remarks to the Author):

Thank you, my comments have been satisfactorily addressed. Although we can't be certain about fatty acids and micronutrient differences between the diets and their potential impact, I think the explanation to readers is reasonable.